# Learning Latent Dynamics for Partially-Observed Chaotic Systems

## Abstract

This paper addresses the data-driven identification of latent Ordinary Differential Equation (ODE) representations of partially-observed dynamical systems, *i.e.* dynamical systems whose some components are never observed, with an emphasis on forecasting applications and long-term asymptotic patterns. Whereas state-of-the-art data-driven approaches rely on an explicit mapping of the observation series onto a latent space, the proposed approach only relies on the definition of an augmented space, higher-dimensional than the manifold spanned by the observed variables, where the dynamics of the observations can be fully described by an ODE. From a numerical point of view, the proposed approach exploits a neural-network ODE representation and the associated variational minimization scheme. Numerical experiments support the relevance of the proposed framework w.r.t. state-of-the-art approaches, including neural ODE schemes, both in terms of short-term forecasting errors and long-term behaviour. We further discuss how the proposed framework relates to Koopman operator theory and Takens' embedding theorem.

## 1 Introduction

Learning the underlying dynamical representation of some observed variables $\mathbf{x}_t \in \mathbb{R}^n$ (where $t \in \{t_0, ..., T\}$ is the temporal sampling time and $n$ the dimension of the observations) is a key challenge in various scientific fields, including control theory, geoscience, fluid dynamics, economics; for applications ranging from system identification to forecasting and assimilation issues (Lai & Wei, 1982; Abarbanel & Lall, 1996; Jeong & Hussain, 1995; Koopmans, 1949).

For fully-observed systems, *i.e.* when the observed variables $\mathbf{x}_t$ relate to some underlying deterministic states $\mathbf{s}_t$ according to a diffeomorphic mapping possibly corrupted by noise, recent advances (Brunton et al., 2016b; Fablet et al., 2018; Nguyen et al., 2019) have shown that one can identify the governing equations of the dynamics of $\mathbf{s}_t$ from a representative dataset of observations $\{\mathbf{x}_{t_i}\}_i$. Unfortunately, when the observed variables $\mathbf{x}_t$ only relate to some but not all the components of underlying states $\mathbf{s}_t$, these approaches can not apply since no ODE or, more generally, no one-to-one mapping defined in the observation space can represent the time evolution of the observations. In this context, Takens's theorem states the conditions under which a delay embedding, formed by lagged versions of the observed variables, guarantees the existence of governing equations in the embedded space (Takens, 1981).

Takens's theorem has motivated a rich literature of machine learning schemes to identify dynamical representations of partially-observed systems using a delay embedding. This comprises both non-parametric schemes based on nearest-neighbors or analogs (Abarbanel, 1996a) as well as parametric schemes which include polynomial representations (Paduart et al., 2010), neural network models (Wan, 1993) and Support Vector Regression (SVR) models (Mattera & Haykin, 1999; Müller et al., 1999). For all these approaches, the identification of the appropriate delay embedding is a critical issue (Abarbanel, 1996b;c).

From a neural network and machine learning perspective, the inference of a latent space, within a State Space Model (SSM) framework, for dynamical systems has motivated a broad literature especially for time series forecasting (Ghahramani & Roweis, 1999; Wang et al., 2006; Mirowski & LeCun, 2009; He et al., 2015; Krishnan et al., 2016). Most of those techniques were introduced in the context of reduced order modeling (ROM) to infer low-dimensional manifolds, where the

dynamics of the observations can be represented. When considering partially-observed systems, these approaches state this issue as the inference of a (non-linear) projection of an input sequence in a latent space where the observations can be modeled. This projection is usually computed in a probabilistic framework using Bayesian filtering techniques. However, recovering the attractor's dynamics using iterative predictions is still an issue for such models since the explicit modeling of latent space as a delay embedding of the observations may limit the expressiveness of the latent states, especially when considering chaotic dynamics.

In this work, we show that we do not need to rely explicitly neither on a delay embedding nor on the learning of an inference model to map the observation series to a latent space. We only assume that we may define an augmented space, higher-dimensional than the manifold spanned by the observed variables, where the dynamics of the observations can be fully described by an ODE. Using neural-network representations for the parametrization of the dynamical model, it amounts to jointly learning the governing ODE and reconstructing the augmented latent states for a given observation dataset using a variational framework. We report experiments on linear and chaotic dynamics, which illustrate the relevance of the proposed framework compared to state-of-the-art approaches. We then further discuss the key features of this framework with respect to state-of-the-art dynamical systems identification tools such as Koopman operator theory (Koopman, 1931).

## 2 BACKGROUND AND RELATED WORK

This section introduces the learning of dynamical representations for partially-observed systems and links this problem to recent advances in machine learning.

Let us consider an **unobserved** state variable $\mathbf{s}$ governed by an autonomous system of $m$ differential equations $\dot{\mathbf{s}}_t = f(\mathbf{s}_t)$. Let us also assume that this system generates a flow $\Phi_{t_i}(\mathbf{s}_{t_0}) = \int_{t_0}^{t_i} f(\mathbf{z}_w)dw \in \mathbb{R}^m$ with trajectories that are asymptotic to a limit-cycle $L$ of dimension $d$ contained in $\mathbb{R}^m$. We further assume that we are provided with a measurement function $\mathcal{H}$ that maps our state variable $\mathbf{s}$ to our observations $\mathbf{x}_t = \mathcal{H}(\mathbf{s}_t) \in \mathbb{R}^n$.

When considering the data-driven identification of a dynamical mapping that governs some observation data, we first need to evaluate whether the dynamics in the observation space can be described using a smooth[1] ODE. Another way to tackle this question is to find the conditions under which the deterministic properties of the unobserved limit-cycle $L$ are preserved in the observation space in $\mathbb{R}^n$ such that one can reliably perform forecasts in the observation space. The general condition under which a mapping $\mathcal{H}$ preserves the topological properties of the initial limit-cycle involves a differential structure. Assuming that $L$ is a smooth compact differential manifold, the topological properties of $L$ are preserved through a mapping $\mathcal{H}$ in $\mathbb{R}^n$ if $\mathcal{H}$ is one-to-one and is an immersion of $L$ in $\mathbb{R}^n$. Under these conditions our observation mapping is called an **embedding** (Sauer et al., 1991).

The simplest example of an embedding involves an identity observation operator $\mathcal{H}$. With this embedding, we have direct access to the state variable $\mathbf{s}$ which is governed by a deterministic ODE. This particular case has been widely studied in the literature. Parametric representations have been for decades the most popular models thanks to their simplicity and interpretability (Paduart et al., 2010), (Brunton et al., 2016b). Recently, these approaches have been enriched by neural network and deep learning schemes (Wiewel et al., 2018), (Raissi et al., 2018). In particular, the link between residual networks (Chen et al., 2018; Ouala et al., 2019) and numerical integration schemes have opened new research avenues for learning extremely accurate dynamical models even from irregularly-sampled training data. These schemes show greater interpretability and forecasting performance for the data-driven representation of systems governed by an ODE, compared with other state-of-the-art neural networks schemes, including Recurrent Neural Networks (RNN) such as LSTM (Long-Short-Term Memory).

However, for a wide range of real-world systems, we are never provided with an observation operator that forms an embedding of the unobserved dynamical system. In such situations, we do not have any guarantee on the existence of a smooth ODE that governs the temporal evolution of our observations. From this point of view, the question of finding an appropriate dynamical representa-

---

[1]The word smooth here stands for continuously differentiable or $\mathcal{C}^1$.

tion of some observed data may not be this straightforward. The fact that our data may come from some unobserved governing equation may restrict the use of the above-mentioned state-of-the-art algorithms. The main difficulty lies in the ability to map observation series to a latent space that provides at least a *one-to-one* mapping between two successive states. From a geometrical point of view, the time delay theorem (Takens, 1981) provides a way to build a latent space that preserves the topological properties of the true (unobserved) dynamics limit-cycle. A generalization of this theorem (Sauer et al., 1991) shows that one can reconstruct topologically similar limit-cycles using any appropriate smooth composition map of the observations. The derivation of a dynamical system from such representations however encounters large disparities since no explicit relationships between the defined phase space and an ODE formulation have been clearly made. Classical state-of-the-art techniques such as polynomial representations (Brunton et al., 2016b) and K-Nearest Neighbors (KNN) (Lguensat et al., 2017) algorithms were proposed but they often fail to achieve both accurate short-term forecasting performance and long-term topologically similar reconstructed limit-cycles (see experiments for an illustration).

We may also point out that the limitation of ODE-based representation in deep learning architecture has also been pointed out recently in (Dupont et al., 2019; Zhang et al., 2019) for classification issues. As ODE-derived trajectories do not intersect, it may limit the ability of neural ODE representations to reach relevant classification performance in a given feature space. To address this issue, (Dupont et al., 2019) and (Zhang et al., 2019) propose to consider an augmented state, simply by augmenting the observed state by a number of zeros to create a high-dimensional space in which an ODE representation can be identified. Such a strategy cannot apply to time series modeling as successive augmented states cannot be forced to zero for some dimensions.

Advances in the inference of latent spaces in state space models was introduced essentially, from a dynamical systems perspective, to retrieve low-dimensional manifolds, where the dynamics of the system evolve. When applied to partially-observed systems, the latent variables are typically inferred from a sequence of observations through a parametric modeling of the posterior distribution as in (He et al., 2015; Krishnan et al., 2016; Chen et al., 2018) or through marginalization with model constraints as in (Ghahramani & Roweis, 1999; Wang et al., 2006) However, such models often fail in accounting for long term patterns (as shown in the experiments). This is due to the fact that the latent space is constrained to be a non-linear projection of a sequence of observations, which limits the expressiveness of the dynamical model. Interestingly, (Mirowski & LeCun, 2009) does not involve the learning of an inference model as the reconstruction of the latent states is solved as gradient-based minimization of the dynamical prior w.r.t. an observation series. However, the dynamical prior relies on an explicit delay embedding as the dynamics of the latent state depend both on the previous latent state and on a delay embedding of the observations.

In this work, we address the identification of a latent embedding, associated with an ODE representation, for partially-observed systems. The core idea of this work is to infer an augmented latent space, governed by an ODE, which fully explain the observed time series and their dynamics. In contrast to previous work (He et al., 2015; Krishnan et al., 2016; Chen et al., 2018; Ghahramani & Roweis, 1999; Wang et al., 2006), we do not exploit either a delay embedding or an explicit modeling of the inference model (i.e., the reconstruction of the latent states given the observed time series). As such, our scheme only involves the selection of the class of ODEs of interest. The expected benefits are as follows: (i) our model ensures the existance of a latent embedding associated with an ODE, which may not be guaranteed when considering a parametric inference model and/or a delay embedding, (ii) our model reduces the complexity of the overall scheme to the complexity of the ODE representation, (iii) our model guarantees the consistency of the reconstructed latent states w.r.t. the learnt ODE. Our experiments suggest that this may be of key importance to jointly address short-term forecasting performance and long-term chaotic patterns.

## 3 LEARNING LATENT REPRESENTATIONS OF PARTIALLY-OBSERVED DYNAMICS

**Augmented latent dynamics**: Let us consider a continuous $m$-dimensional dynamical system $\mathbf{s}_t$ governed by an autonomous ODE $\dot{\mathbf{s}}_t = f(\mathbf{s}_t)$ with $\Phi_t$ the corresponding flow $\Phi_t(\mathbf{s}_{t_0}) = \int_{t_0}^{t} f(\mathbf{s}_w)dw$. We assume the state $\mathbf{s}_t$ to be never fully-observed. Formally, we can define an obser-

vation function $\mathcal{H} : \mathbb{R}^m \longrightarrow \mathbb{R}^n$ such that the observation $\mathbf{x}_t$ follows $\mathbf{x}_t = \mathcal{H}(\mathbf{s}_t)$. An example of interest comprises the situation when one observes only the $n$ first variables of $\mathbf{s}_t$.

Our goal is to derive an ODE representation of $\mathbf{x}_t \in \mathbb{R}^n$. However, the key question arising here is the extent to which the dynamics expressed in the observations space, reflect the true underlying dynamics in $\mathbb{R}^m$, and consequently, the conditions on $\mathcal{H}$ under which the predictable deterministic dynamical behavior of the hidden states is still predictable in the observations space. To illustrate this issue, we may consider a linear dynamical system in the complex domain governed by the following linear ODE:

$$\begin{cases} \dot{\mathbf{s}}_t = \alpha \mathbf{s}_t \\ \mathbf{s}_{t_0} = \mathbf{s}_0 \end{cases} \tag{1}$$

with $\mathbf{s} \in \mathbb{C}$ a state variable and $\alpha \in \mathbb{C}$ a complex imaginary number. The solution of this problem is

$$\mathbf{s}_t = K e^{\alpha t} \tag{2}$$

with $K$ a constant depending on $\mathbf{s}_0$. Let us assume now that we are only provided with the real part as direct measurements of the unobserved state *i.e.* $\mathcal{H}(.) = Real(.) : \mathbf{x}_t = Real(\mathbf{s}_t)$.

**Proposition 1** : *The flow of an ODE cannot represent the time evolution of $\mathbf{x}_t$.*

The proof of the proposition is given in the appendix. The intuition exemplified for the 2-dimensional system given by (1) is as follows. If we assume that we are only provided the real part as direct measurements $\mathbf{x}_t \in \mathbb{R}$ of the true states $\mathbf{s}_t$, no smooth autonomous ODE model in the scalar observation space can describe the trajectories of the observations as the mapping between two observations is not one-to-one. For example, assuming that $\mathbf{s}_{t_0}$ and $\mathbf{s}_{t_1}$ correspond to two states that have the same real part but distinct imaginary parts, the associated observed states are equal $\mathbf{x}_{t_0} = \mathbf{x}_{t_1}$. However, the time evolution of the states $\mathbf{s}_{t_0}$ and $\mathbf{s}_{t_1}$ differ if they have different imaginary parts. The observed states $\mathbf{x}_{t_0+\delta}$ and $\mathbf{x}_{t_1+\delta}$ are then no longer equal after any time increment $\delta$. As a consequence, a given observation may have more than one future state and this behavior cannot be represented by a smooth ODE in the observation space. Hence, the application of an ODE mapping such as (Chen et al., 2018) and (Fablet et al., 2018) for such observations will lead to poor forecasting performance. From a naive Neural networks point of view, fitting such a model will most likely force the forecasting into an equilibrium point since we may iteratively match the same inputs to different output predictions. This simple example emphasizes the need for considering augmented latent states.

In this context, Takens's theorem guarantees the existence of an augmented space, in which a one-to-one mapping exists between successive time steps (Takens, 1981). More precisely, for a given observation operator $\mathcal{H}$ of a deterministic underlying dynamical system that governs $\mathbf{s}_t$, one can define a delay embedding from an observation sequence, *i.e.* $\{\mathbf{x}_{t_0-a\delta}, \mathbf{x}_{t_0}, \dots, \mathbf{x}_{t_0+b\delta}\}$ with $\delta$ the time delay and $a, b$ two natural numbers. An ODE representation in a latent space can than be formulated according to (Chen et al., 2018) by sampling the latent states from a parametrized posterior distribution given the delay embedding time series. Here, we aim to identify an augmented latent space without resorting to an explicit mapping from the observed series to the latent space, such that the latent dynamics are governed by a smooth ODE and can be mapped to the observations. Formally, let us introduce the following notations. Let $\mathbf{u}_t \in \mathbb{R}^{d_E}$ define a $d_E$-dimensional augmented latent state formed by an observed component[2] $\mathbf{x}_t$ and an unobserved latent component referred to as $\mathbf{z}_t$.

$$\mathbf{u}_t^T = [\mathbf{x}_t^T, \mathbf{z}_t^T] \tag{3}$$

Assuming that the augmented latent state $\mathbf{u}_t$ is governed by an ODE leads to the the following state space model :

$$\begin{cases} \dot{\mathbf{u}}_t = f_\theta(\mathbf{u}_t) \\ \mathbf{x}_t = G(\mathbf{u}_t) \end{cases} \tag{4}$$

where the dynamical operator $f_\theta$ belongs to a family of smooth operators (in order to guarantee uniqueness (Coddington & Levinson, 1955)) parametrized by $\theta$. $G$ is a projection matrix that satisfies $\mathbf{x}_t = G(\mathbf{u}_t)$. It means that operator $G$ only returns the observed component of the augmented

---

[2]For simplicity, the observed component is considered in the equations as the direct observations. However, in general, when considering Reduced Order Models (ROMs), the observed component can be seen as a low dimensional projection of the observations. Please refer to the appendix for a similar derivation of the equations in the case of reduced order modeling.

latent state. Compared with previous work, the key difference lies in the definition of the augmented latent space, where only the observed component is explicitly stated as a function of observation $\mathbf{x}_t$. Similarly to neural ODE frameworks (Chen et al., 2018; Fablet et al., 2018), we consider a neural-network representation with Lipschitz nonlinearities and finite weights for ODE operator $f_\theta$. As detailed in the next sections, we address the identification of operator $f_\theta$ from a dataset of observed state series $\{\mathbf{x}_0, \ldots, \mathbf{x}_T\}$ as well as the exploitation of the identified latent dynamics for the forecasting of the time evolution of the observed states, for instance unobserved future states $\{\mathbf{x}_{T+1}, \ldots, \mathbf{x}_{T+N}\}$. Both issues involve the joint inference of the unobserved component $\mathbf{z}_t$ of the augmented latent states. This formulation generalizes to reduced-order modeling (ROM) (Champion et al., 2019), where observation $\mathbf{x}_t$ fully derives from a lower-dimensional variable. We let the reader refer to the appendix for the associated generalization of the proposed framework.

**Learning scheme**: Given an observation time series $\{\mathbf{x}_0, \ldots, \mathbf{x}_T\}$ we aim to identify the state-space model defined by (4), which amounts to learning the parameters $\theta$ of the dynamical operator $f_\theta$. However, as the component $\mathbf{z}_t$ of the augmented state $\mathbf{u}_t$ is never observed, this identification requires the joint optimization of the model parameters $\theta$ as well as of the hidden component $\mathbf{z}_t$. Formally, this problem is stated as the following minimization of the forecasting error on observed variables:

$$\hat{\theta} = \arg\min_\theta \min_{\{\mathbf{z}_t\}_t} \sum_{t=1}^{T} \|\mathbf{x}_t - G(\Phi_{\theta,t}(\mathbf{u}_{t-1}))\|^2$$

$$\text{Subject to} \begin{cases} \mathbf{u}_t &= \Phi_{\theta,t}(\mathbf{u}_{t-1}) \\ G(\mathbf{u}_t) &= \mathbf{x}_t \end{cases} \tag{5}$$

with $\Phi_{\theta,t}$ the one-step-ahead diffeomorphic flow associated with operator $f_\theta$ such that:

$$\Phi_{\theta,t}(\mathbf{u}_{t-1}) = \mathbf{u}_{t-1} + \int_{t-1}^{t} f_\theta(\mathbf{u}_w)dw$$

In (5), the loss to be minimized involves the one-step-ahead forecasting error for the observed variable $\mathbf{x}_t$. The constraints state that the augmented latent state sequence $\{\mathbf{u}_0, \ldots, \mathbf{u}_T\}$ defined by (3) should be a solution of the ODE (4). Here, we numerically minimize the equivalent formulation:

$$\min_\theta \min_{\{\mathbf{z}_t\}_t} \sum_{t=1}^{T} \|\mathbf{x}_t - G(\Phi_{\theta,t}(\mathbf{u}_{t-1}))\|^2 + \lambda\|\mathbf{u}_t - \Phi_{\theta,t}(\mathbf{u}_{t-1})\|^2 \tag{6}$$

where $\mathbf{u}_t^T = [\mathbf{x}_t^T, \mathbf{z}_t^T]$ and $\lambda$ a weighting parameter. The term $\|\mathbf{u}_t - \Phi_{\theta,t}(\mathbf{u}_{t-1})\|^2$ may be regarded as a regularization term such that the inference of the unobserved component $\mathbf{z}_t$ of the augmented state $\mathbf{u}_t$ is not solved independently for each time step.

Using a neural-network parametrization for the ODE operator $f_\theta$, the corresponding forecasting flow $\Phi_{\theta,t}$ is also stated as a neural network based on a numerical integration scheme formulation (typically a $4^{th}$-order Runge-Kutta scheme). This architecture, very much similar to a ResNet (He et al., 2015), allows very accurate identification of ODE models (Fablet et al., 2018; Ouala et al., 2019). Hence, for a given observation sequence $\{\mathbf{x}_0, \ldots, \mathbf{x}_T\}$, we minimize (6) jointly w.r.t. $\theta$ and unobserved variables $\{\mathbf{z}_0, \ldots, \mathbf{z}_T\}$.

**Application to forecasting**: We also apply the proposed framework to the forecasting of the observed states $\mathbf{x}_t$. Given a trained latent dynamical model (4), forecasting future states for $\mathbf{x}_t$ relies on the forecasting of the entire augmented latent state $\mathbf{u}_t$. The latter amounts to determining an initial condition of the unobserved component $\mathbf{z}_t$ and performing a numerical integration of the trained ODE (4).

Let us denote by $\mathbf{x}_t^n$, $t \in \{t_0, ..., T\}$ a new series of observed states. We aim to forecast future states $\mathbf{x}_t^n$, $t \in \{T+1, ..., T+\delta T\}$. Following (6), we infer the unobserved component $\hat{\mathbf{z}}_T$ of the augmented state $\mathbf{u}_T^n$ at time $T$ from the following minimization:

$$\hat{\mathbf{z}}_T^n = \arg\min_{\mathbf{z}_T^n} \min_{\{\mathbf{z}_t^n\}_{t<T}} \sum_{t=t_0}^{T} \|\mathbf{x}_t^n - G(\Phi_{\theta,t}(\mathbf{u}_{t-1}^n))\|^2 + \lambda\|\mathbf{u}_t^n - \Phi_{\theta,t}(\mathbf{u}_{t-1}^n)\|^2 \tag{7}$$

Here, we only minimize w.r.t. latent variables $\{\mathbf{z}_t^n\}$ given the trained forecasting operator $\Phi_{\theta,t}$. This minimization relates to a variational assimilation issue with partially-observed states and known dynamical and observation operators (Lynch & Huang, 2010). Similarly to the learning step, we benefit from the neural-network parameterization of operator $\Phi_{\theta,t}$ and from the associated automatic differentiation tool to compute the solution of the above minimization using a gradient descent.

We may consider different initialization strategies for this minimization problem. Besides a simple random initialization, we may benefit from the information gained on the manifold spanned by the unobserved components during the training stage. The basic idea comes to assume that the training dataset is likely to comprise state trajectories which are similar to the new ones. As the training step embeds the inference of the whole latent state sequence, we may pick as initialization for minimization (7) the inferred augmented latent state in the training dataset which leads to the observed state trajectory that is the most similar (in the sense of the L2 norm) to the new observed sequence $\mathbf{x}_t^n$. The interest of this initialization scheme is two-fold: (i) speeding-up the convergence of minimization (7) as we expect to be closer to the minimum; (ii) considering an initial condition which is in the basin of attraction of the reconstructed limit-cycle. The latter may be critical as we cannot guarantee that the learnt model does not involve other limit-cycles than the ones truly revealed by the training dataset, which may lead to a convergence to a local and poorly relevant minimum.

## 4 NUMERICAL EXPERIMENTS

In this section, we report numerical experiments to illustrate the key features of proposed framework. We consider three case-studies: a linear ODE case-study; a chaotic system, namely Lorenz-63 dynamics, and real upper ocean data.

**Application to a linear ODE**: In order to illustrate the key principles of the proposed framework, we consider the following linear ODE in the complex domain:

$$\begin{cases} \dot{\mathbf{s}}_t = \alpha \mathbf{s}_t \\ \mathbf{s}_{t_0} = \mathbf{s}_0 \end{cases} \tag{8}$$

with $\alpha = -0.1 - 0.5j$, $j^2 = -1$ and $\mathbf{s}_0 = 0.5$. As $\alpha \in \mathbb{C}$ with $Real(\alpha) < 0$ and $\mathbf{s}_0 \neq 0$, the solution of this ODE is an ellipse in the complex plane (Fig. 1).

As observations, we consider the real part of the underlying state, *i.e.* the observation function $\mathcal{H} : \mathbb{C} \longrightarrow \mathbb{R}$ is given by $\mathbf{x}_t = Real(\mathbf{s}_t)$. This is a typical example, where the mapping between two successive observations is not a one-to-one mapping since all the states that have the same real part lead to the same observation. As explained in section 3, one cannot identify an autonomous ODE model that will reproduce the dynamical behavior of the observations in the observations space.

We apply the proposed framework to this toy example. We consider a 2-dimensional augmented state $\mathbf{u}_t = [\mathbf{x}_t, \mathbf{z}_t^1]$. As neural-network parametrization for operator $f_\theta$, we consider a neural network with a single linear fully-connected layer. We use an observation series of 10000 time steps as training data. As illustrated in Fig.1, given the same initial condition over the observable state, the inferred latent state dynamics, though different from the true ones, depicts a similar spiral pattern. This result is in agreement with the geometrical reconstruction techniques (Takens, 1981) of the latent dynamics up to a diffeomorphic mapping. Overall, our model learns a dynamical behavior similar to the true model represented by an elliptic transient and an equilibrium point limit-set. Furthermore, the projection of the augmented latent space and the true solution of Eq. (8) in the real axis illustrate the relevance of the proposed framework in forecasting the observations dynamics (mean square error $< 1E - 6$).

**Lorenz-63 dynamics**: Lorenz-63 dynamical system is a 3-dimensional model that involves, under some specific parametrizations (Lorenz, 1963), chaotic dynamics with a strange attractor. We simulate chaotic Lorenz-63 state sequences with the same model parameters as proposed in (Lorenz, 1963) using the LOSDA ODE solver (Hindmarsh, 1983) with an integration step of 0.01. We assume that only the first Lorenz-63 variable is observed $\mathbf{x}_t = \mathbf{s}_{t,1}$. We apply the proposed framework to this experimental setting using a training sequence of 4000 time-steps.

For benchmarking purposes, we perform a quantitative comparison with state-of-the-art approaches using delay embedding representations (Takens, 1981). The parameters of the delay embedding

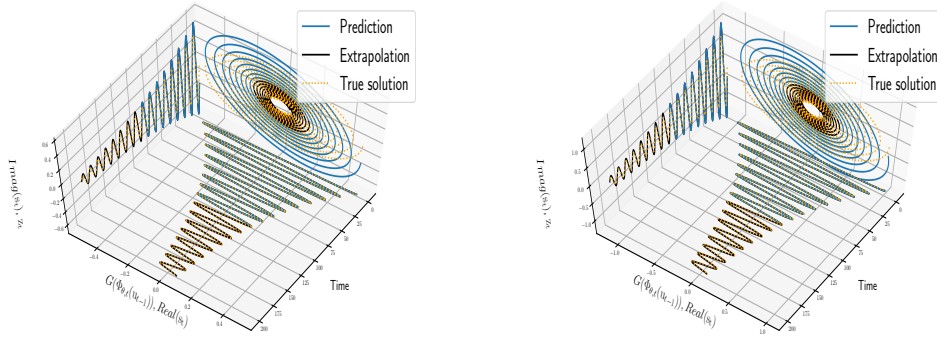

Figure 1: **Illustration for a 2-dimensional linear ODE**: Forecasted augmented latent space with respect to the true states given (left) the same initial condition as the training sequence and (right) a different initial condition. Given the initial condition we illustrate both the prediction (forecast up to the end of the training time) of the trained model and the extrapolation (forecast beyond the training time) performances with respect to the true trajectory .The projection of the solutions in the real plane illustrates the forecasting of the observations.

representation, namely the lag $\tau$ and the dimension $d_E$ of the augmented space were computed using state-of-the-art techniques. Specifically, the lag parameter was computed using both the mutual information and correlation techniques (Abarbanel, 1996b), respectively denoted as $\tau_{MI}$ and $\tau_{Corr}$. Regarding the dimension of the embedding representation, we used the Takens embedding condition $d_E = 2d + 1$ with $d$ the dimension of the hidden limit-cycle. The delay embedding dimension was also computed using the False Nearest Neighbors (FNN) method (Abarbanel, 1996c). We also tested arbitrary parameters for the delay embedding dimension. Given the delay embedding representation, we tested two state-of-the-art data-driven representations of the dynamics. The Analog Forecasting technique (AF) which is based on the nearest neighbours algorithm (Lguensat et al., 2017) and the Sparse Regression (SR) method on a second order polynomial representation of the delay embedding states. Regarding deep learning models, we compare our method to a stacked Bidirectional LSTM (RNN) and to the Latent-ODE model as proposed in (Chen et al., 2018). Finally, the proposed framework, referred to as Neural embedding for Dynamical Systems (NbedDyn) was tested for different dimensions of the augmented state space, namely from 3 to 6 (please refer to the apendix for details on the neural networks based architectures)[3].

Fig. 2 illustrates the learning process for the latent space from the initialization to the last training epoch. We also report the analysis of short-term forecasting performance as well as the long-term asymptotic behavior characterized by the largest Lyapunov exponent of the benchmarked models in Tab 1. The proposed model leads to significant improvements in terms of short term forecasting performance with respect to the other approaches. Surprisingly, The Latent-ODE and RNN models lead to the poorest performance both in terms of forecasting error and asymptotic behavior. This is mainly due, in the Latent-ODE case, to the fact that the latent space is seen as a non linear projection of the observed variables through the optimization of the ELBO loss (Krishnan et al., 2016). By contrast, our latent embedding formulation optimizes the latent states to forecast the observed variables which explicitly constrain the latent space to be an embedding of the true underlying dynamics. The RNN model in the other hand converges to a periodic solution (please refer to the appendix for forecasting figures) with still a poor short term forcasting performances. Overall, this results on deep learning models suggest that one should use such tools with care to guarantee satisfying the specifications of the underlying system.The SR model seems to lead to better short term forecast (using ad hoc parameters ($\tau = 6$, $d_E = 3$), however, it does not capture well the chaotic

---

[3]The results of the neural networks based models were averaged over 5 runs except for NbedDyn $d_E = 4$ and $d_E = 5$ where we only launched 3 runs. We will include other runs for the final version of the manuscript

| | Model | | $t_0 + h$ | $t_0 + 4h$ | $\lambda_1$ |
|---|---|---|---|---|---|
| AF | $\tau_{MI}=16$ | $d_E(FNN)=3$ | $5.6E-3$ | $1.3E-2$ | $0.85$ |
| | $\tau_{MI}=16$ | $d_E(Takens)=6$ | $9.9E-3$ | $2.4E-2$ | $NaN$ |
| | $\tau_{Corr}=27$ | $d_E(FNN)=3$ | $8.9E-3$ | $2.3E-2$ | $12.35$ |
| | $\tau_{Corr}=27$ | $d_E(Takens)=6$ | $8.5E-3$ | $1.9E-2$ | $NaN$ |
| | $\tau=6$ | $d_E=3$ | $8.0E-4$ | $9.0E-4$ | $0.87$ |
| | $\tau=10$ | $d_E=3$ | $2.1E-3$ | $4.9E-3$ | $0.60$ |
| SR | $\tau_{MI}=16$ | $d_E(FNN)=3$ | $7.8E-2$ | $2.5E-1$ | $0.12$ |
| | $\tau_{MI}=16$ | $d_E(Takens)=6$ | $4.5E-2$ | $1.7E-1$ | $NaN$ |
| | $\tau_{Corr}=27$ | $d_E(\text{FNN})=3$ | $1.4E-1$ | $4.6E-1$ | $NaN$ |
| | $\tau_{Corr}=27$ | $d_E(\text{Takens})=6$ | $2.1E-1$ | $8.4E-1$ | $NaN$ |
| | $\tau=6$ | $d_E=3$ | $7.6E-3$ | $7.4E-3$ | $NaN$ |
| | $\tau=10$ | $d_E=3$ | $2.5E-2$ | $5.7E-2$ | $0.2535$ |
| Latent-ODE | | | $6.9E-2 \pm 2.9E-2$ | $1.5E-1 \pm 3E-2$ | $NaN$ |
| RNN | | | $6.9E-2 \pm 4.6E-2$ | $1.5E-1 \pm 1.1E-1$ | $-6.79 \pm 0.0$ |
| NbedDyn | $d_E=3$ | | $3.2E-4 \pm 1.3E-4$ | $1.7E-3 \pm 7.5E-4$ | $0.81 \pm 0.09$ |
| | $d_E=4$ | | $1.3E-4 \pm 5.2E-5$ | $7.3E-4 \pm 2.2E-4$ | $0.82 \pm 0.06$ |
| | $d_E=5$ | | $3.8E-4 \pm 7.4E-4$ | $2.0E-3 \pm 3.4E-4$ | $0.80 \pm 0.02$ |
| | $d_E=6$ | | $3.7E-4 \pm 2.8E-4$ | $2.0E-3 \pm 1.7E-3$ | $0.92 \pm 0.02$ |
| | $d_E=6$ (Best) | | $9.1E-5$ | $4.7E-4$ | |

Table 1: ***Forecasting performance on the test set of data-driven models for Lorenz-63 dynamics where only the first variable is observed***: first two columns : mean RMSE for different forecasting time steps, third column : largest Lyapunov exponent of a predicted series of length of 10000 time-steps (The true largest Lyapunov exponent of the Lorenz 63 model is 0.91 (Sprott, 2003)).

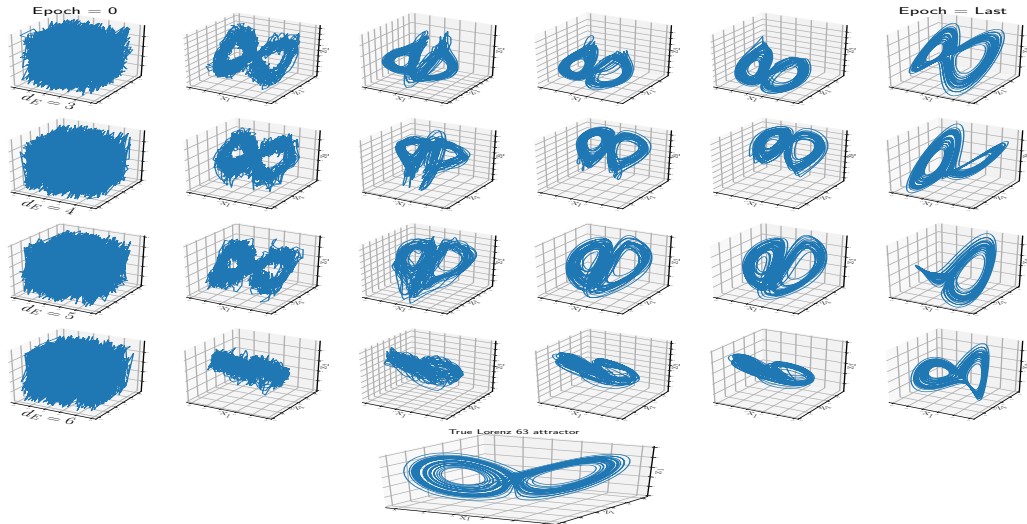

Figure 2: ***Evolution of the learnt latent space***: starting from a random initialization of the augmented states $\mathbf{z}_i$, the latent space is optimized according to the minimization of Eq. (6) to form a limit-cycle similar to the true Lorenz 63 attractor. We depict 3-dimensional projections of the learnt latent space for the proposed model with different embedding dimensions from $d_E = 3$ to $d_E = 6$.

patterns, which are associated to a positive largest Lyapunov exponent. This may suggest the combination of the SR model and a delay embedding may require additional investigation as a good geometrical reconstruction of the phase space as stated in Takens' theorem does not guarantee the existence of a parametric ODE model based on the corresponding delay embedding variables. Better performance is reported using an analog forecasting approach. The performance however greatly varies depending on the considered definition of the delay embedding. Using ad hoc parameters ($\tau = 6$, $d_E = 3$), one may retrieve the expected long-term chaotic behavior ($\lambda_1 = 0.87$) with a relatively low short-term forecasting error ($8.0e - 4$ for a one-step-ahead forecast). When considering the proposed model, we report for all parametrizations, augmented space dimensions from 3 to 6, performance at least in the same range as the best analog forecasting setting. Besides, when

increasing the dimension of the augmented space, we significantly decrease short-term forecasting errors ($< 1.e - 4$ for a one-step-ahead forecast when considering the best fit for $d_E = 6$, i.e. one order of magnitude compared to the best benchmark model) while keeping an appropriate chaotic long-term pattern ($\lambda_1 = 0.92$).

**Modeling Sea Level Anomaly (SLA)**: The data driven identification of dynamical representations of real data is an extremely difficult task especially when the underlying processes involve non stable behaviors such as chaotic attractors. This is mainly due to the fact that we do not have any exact knowledge of the closed form of the equations governing the temporal evolution of our variables. Furthermore, the measured quantity may depend on other unobserved variables which makes the exploitation of data-driven techniques highly challenging.

In this context, we report an application to SLA (Sea Level Anomaly) dynamics, which relate to upper ocean dynamics and are monitored by satellite altimeters (Calmant et al., 2008). Sea surface dynamics are chaotic and clearly involve latent processes, typically subsurface and atmospheric processes. The dataset used in our experiments is a SLA time series obtained using the WMOP product (Juza et al., 2016). The spatial resolution of our data is a $0.05°$ and the temporal resolution $h = 1$ day. We use the data from January 2009 to December 2014 as training data and we tested our approach on the last month of the year 2014. The considered region is located on south Mallorca ($2.5°E$ to $4.25°E$, $37.25°N$ to $39.5°N$). Finally, and in order to identify a ROM, our observations are mapped to a low dimensional space using a projection matrix defined offline using a PCA as follow : $\mathbf{r}_t = \mathcal{M}(\mathbf{x}_t) \in \mathbb{R}^k$ with $k = 15$ which amounts to capture 92% of the total variance. The proposed framework is applied simply by replacing $\mathbf{x}_t$ by the PCA components $\mathbf{r}_t$. Please refere to the appendix for a generalization of the proposed approache in the case of ROM.

We report forecasting performance for our model and include a comparison with analog methods (AF), Sparse regression (SR), LSTM (RNN) and a neural ODE setting (Latent-ODE) in Tab. 2 (The results of the neural networks based models were averaged over 5 runs). Regarding the proposed NbedDyn model we consider an augmented latent space with $d_E = 60$. Our model clearly outperforms the three benchmarked schemes with a very significant gain for the forecasting performance at one day (relative gain greater than 90 %) and two days (relative gain greater than 90 %). For

| Model | | $t_0 + h$ | $t_0 + 2h$ | $t_0 + 4h$ |
|---|---|---|---|---|
| AF | RMSE | 0.036 | 0.049 | 0.067 |
| | Corr | 98.93% | 96.97% | 93.99% |
| SR | RMSE | 0.014 | 0.021 | xx 0.037 |
| | Corr | 99.42% | 97.63% | 90.91% |
| Latent-ODE | RMSE | $0.030 \pm 0.05$ | $0.031 \pm 0.031$ | $0.040 \pm 0.040$ |
| | Corr | $98.20\% \pm 0.39\%$ | $97.39\% \pm 0.36\%$ | $93.42\% \pm 0.55\%$ |
| RNN | RMSE | $0.026 \pm 0.003$ | $0.038 \pm 0.007$ | $0.053 \pm 0.016$ |
| | Corr | $98.36\% \pm 0.40\%$ | $95.29\% \pm 1.73\%$ | $74.97\% \pm 5.75\%$ |
| NbedDynZERO | RMSE | $0.016 \pm 0.0$ | $0.023 \pm 0.0$ | $0.038 \pm 0.0$ |
| | Corr | $99.44\% \pm 0.0\%$ | $97.71\% \pm 0.0\%$ | $91.18\% \pm 0.0\%$ |
| NbedDyn | RMSE | $0.002 \pm 0.0003$ | $0.006 \pm 0.001$ | $0.020 \pm 0.004$ |
| | Corr | $99.99\% \pm 0.0017\%$ % | $99.91\% \pm 0.01\%$ | $99.01\% \pm 0.04\%$ |

Table 2: *SLA Forecasting performance on the test set of data-driven models*: RMSE and correlation coefficients for different forecasting time steps.

a 4-day-ahead forecasting, our model still outperforms the other ones though the gain is lower (relative gain of 40%). Finally, and in order to illustrate the influence of adding extra dimensions to define an augmented latent space on real data, we also tested the proposed NbedDyn model directly on the PCA space ($d_E = k = 15$) this model is referred to as NbedDynZERO and the influence of the latent components is clear from the results in Tab. 2 witch allows a relative gain up to 90 % with respect to the same model directly on the PCA space. We let the reader refer to the Supplementary Material for a more detailed analysis of these experiments, including visual comparisons of the forecasts.

## 5 Discussion

In this work, we address the data-driven identification of latent dynamics for systems which are only partially observed, *i.e.* when some components of the system of interest are never observed. The reported forecasting performance for Lorenz-63 dynamics is in line with the forecasting performance of state-of-the-art learning-based approaches for a noise-free and fully-observed setting. This is of key interest for real-world applications, where observing systems most often monitor only some components of the underlying dynamics. As a typical example, the SLA forecasting experiment

clearly motivates the proposed framework in the context of ocean dynamics for which neither in situ nor satellite observing systems can provide direct observations for all state variables (e.g., subsurface velocities, fine-scale sea surface currents).

We may also further discuss how the proposed framework relates to state-of-the-art dynamical system theory approaches. Most of these approaches rely on delay embedding, as Takens's theorem states the existence of a delay embedding in which the topological properties of the hidden dynamical system are equivalent to those of the true systems up to a diffeomorphic mapping. Hence, state-of-the-art approaches typically combine the selection of a delay embedding representation within classic regression models to represent the one-step-ahead mapping in the considered embedding. Here, we consider latent dynamics governed by an unknown ODE (4) but we do not explicitly state the latent space. This is however implicit in our forecasting framework. By construction, the considered forecasting model relies on the integration of the learnt ODE (4) from an initial condition given as the solution of minimization (7). Let us consider the following embedding $\psi$ such that:

$$\psi\left(\{\mathbf{x}_t\}_{t_0:T}\right) = \arg\min_{\mathbf{u}_T} \min_{\{\mathbf{u}_t\}_{t<T}} \sum_{t=1}^{T} \|\mathbf{x}_t - G\left(\Phi_{\theta,t}\left(\mathbf{u}_{t-1}\right)\right)\|^2 + \lambda\|\mathbf{u}_t - \Phi_{\theta,t}(\mathbf{u}_{t-1})\|^2 \quad (9)$$

Given this embedding, the resulting one-step-ahead forecasting for the observed variable may written as:

$$\mathbf{x}_{T+1} = G\left(\Phi_{\theta,t}\left(\psi\left(\{\mathbf{x}_t\}_{t=t_0:T}\right)\right)\right) \quad (10)$$

Hence, $\psi$ defines a delay embedding representation implicitly stated through minimization (7). In this embedding, the dynamics of the observed system $\mathbf{x}$ is governed by the composition of observation operator $G$ and forecasting operator $\Phi_{\theta,t}$. Regarding the literature on Koopman operator theory, most approaches rely on the explicit identification of eigenfunctions and eigenvalues of the Koopman operator (Koopman, 1931; Brunton et al., 2016a; Tu et al., 2014). Our framework relates to the identification of the infinitesimal generator $f_\theta$ of the one-parameter subgroup defined by Koopman operator through the ODE representation (4). By construction, the Koopman operator associated with the identified operator $f_{\hat{\theta}}$ is also diagonalizable, such that the identification of infinitesimal generator $f_{\hat{\theta}}$ provides an implicit decomposition of the Koopman operator of the underlying and unknown dynamical system onto the eigenbasis of the learnt latent dynamics governed by ODE (4).

Future work will further explore methodological aspects, especially the application to high-dimensional and stochastic systems. In the considered framework, when considering ROM of the SLA experiment, the projection from the observations space to the low dimensional space was carried using simple PCA projection. Although for the geociences community, using PCA to reduce the dimensionality is motivated by the Galerkin derivation of reduced order models from complex high dimensional governing partial differential equations (Holmes et al., 2012), using auto-encoders have shown promising results in discovering optimal coordinates when trained jointly with a dynamical system. The combination of the proposed framework with the variational setting considered in the Latent-ODE model (Chen et al., 2018) also appears as an interesting direction for future work. The extension to stochastic systems through the identification of a Stochastic ODE is also of key interest, for instance for future applications of the proposed framework to geophysical random flows, especially to the simulation and forecasting of ocean-atmosphere dynamics in which stochastic components naturally arise (Chapron et al., 2018).

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

APPENDIX

## A    PROOF OF PROPOSITION 1

This proposition can be easily extended to any observation function that doesn't form an embedding of the initial unobserved ODE. However, for the sake of simplicity, we will consider the example given in Eq. (1).

Lets suppose a a smooth ODE in the observation space that governs the time evolution of $\mathbf{x}$ from Eq. (1).

$$\begin{cases} \dot{\mathbf{x}}_t = f(\mathbf{x}_t) \\ \mathbf{x}_{t_0} = \mathbf{x}_0 \end{cases} \tag{11}$$

This ODE generates a flow $\mathbf{x}_t = \Psi_t(\mathbf{x}_0)$.

Since our observation operator is not one-to-one, we can assume the existence of some $\hat{t}, t_1, t_2$ where $Real(\Phi_{\hat{t}}(\mathbf{s}_{t_1})) = Real(\Phi_{\hat{t}}(\mathbf{s}_{t_2}))$ with $Real(\mathbf{s}_{t_1}) \neq Real(\mathbf{s}_{t_2})$ ($\Phi$ is the flow generated by the unobserved ODE illustrated in Eq. (1)). Projecting this equality to the observation space leads to : $\Psi_{\hat{t}}(\mathbf{x}_{t_1}) = \Psi_{\hat{t}}(\mathbf{x}_{t_2})$ with $\mathbf{x}_{t_1} \neq \mathbf{x}_{t_2}$.

Since the above ODE is smooth (or continuously differentiable), we can show that $f$ is locally Lipschitz on any interval containing $t_0$ (Sohrab, 2003) which garentees by Picard's Existence Theorem the existance of a unique solution (Coddington & Levinson, 1955). Formally, for the times $\hat{t}, t_1, t_2$, $\Psi_{\hat{t}}(\mathbf{x}_{t_1}) = \Psi_{\hat{t}}(\mathbf{x}_{t_2})$ if and only if $\mathbf{x}_{t_1} = \mathbf{x}_{t_2}$. This contradicts the assumption that $\mathbf{x}_{t_1} \neq \mathbf{x}_{t_2}$ and thus, there is no existence of a $\hat{t}$ such that $Real(\Phi_{\hat{t}}(\mathbf{s}_{t_1})) = Real(\Phi_{\hat{t}}(\mathbf{s}_{t_2}))$ with $Real(\mathbf{s}_{t_1}) \neq Real(\mathbf{s}_{t_2})$.

## B    AUGMENTED REDUCED ORDER MODELING

With a view to providing a general formulation which also accounts for reduced-order modeling (ROM) (Champion et al., 2019), we assume that the observations $\mathbf{x}_t$ are fully characterized by a lower-dimensional state $\mathcal{M}(\mathbf{x}_t) \in \mathbb{R}^k$, with $\mathcal{M}$ a global coordinates chart, mapping each observation to its coordinates $\mathcal{M}(\mathbf{x}_t) \in \mathbb{R}^k$. Typical examples for operator $\mathcal{M}$ includes the identity as in the equations of section 3. In ROM, $\mathcal{M}$ may include PCA projections (Benner et al., 2005) for Galerkin derivation of ROM for complex high dimensional partial differential equations (Rowley et al., 2004). Formally, the augmented state space is formulated as :

$$\mathbf{u}_t{}^T = [\mathcal{M}(\mathbf{x}_t)^T, \mathbf{z}_t^T] \tag{12}$$

with the corresponding SSM :

$$\begin{cases} \dot{\mathbf{u}}_t = f_\theta(\mathbf{u}_t) \\ \mathbf{x}_t = \mathcal{M}^{-1}(G(\mathbf{u}_t)) \end{cases} \tag{13}$$

Based on the above formulation, we can state the optimization problem stated in Eq. (5) as:

$$\hat{\theta} = \arg \min_\theta \min_{\{\mathbf{z}_t\}_t} \sum_{t=1}^T \|\mathbf{x}_t - \mathcal{M}^{-1}(G(\Phi_{\theta,t}(\mathbf{u}_{t-1}))) \|^2$$

$$\text{Subject to} \begin{cases} \mathbf{u}_t & = & \Phi_{\theta,t}(\mathbf{u}_{t-1}) \\ \mathcal{M}^{-1}(G(\mathbf{u}_t)) & = & \mathbf{x}_t \end{cases} \tag{14}$$

with the corresponding non constrained formulation :

$$\min_\theta \min_{\{\mathbf{z}_t\}_t} \sum_{t=1}^T \|\mathbf{x}_t - \mathcal{M}^{-1}(G(\Phi_{\theta,t}(\mathbf{u}_{t-1}))) \|^2 + \lambda \|\mathbf{u}_t - \Phi_{\theta,t}(\mathbf{u}_{t-1})\|^2 \tag{15}$$

Finally, when considering forecasting applications of new observations, we can restate the equation 7 in the context of ROM as follow :

$$\hat{\mathbf{z}}_T^n = \arg\min_{\mathbf{z}_T^n} \ \min_{\{\mathbf{z}_t^n\}_{t<T}} \sum_{t=T+1}^{T+\delta T} \|\mathbf{x}_t^n - \mathcal{M}^{-1}(G\left(\Phi_{\theta,t}\left(\mathbf{u}_{t-1}^n\right)\right))\|^2 + \lambda\|\mathbf{u}_t^n - \Phi_{\theta,t}(\mathbf{u}_{t-1}^n)\|^2 \quad (16)$$

## C  LIST OF SYMBOLS

**s**        Unobserved state governing the underlying dynamics in $\mathbb{R}^m$

**x**        Observed component of **s** in $\mathbb{R}^n$, usually related to some components of **s**

$\mathcal{M}$        In the case of ROM (**x** high dimensional), $\mathcal{M}$ is a projection operator from $\mathbb{R}^n$ to $\mathbb{R}^k$ with $n >> k$

**u**        Augmented latent space of dimension $\mathbb{R}^{d_E}$ which consists on a concatenation of an observed components ($\mathcal{M}(\mathbf{x})$) and a latent components **z**

**z**        Latent component concatenated to $\mathcal{M}(\mathbf{x})$ to form the augmented state **u**

$f(.)$        Unknown dynamical system governing the unobserved state **s**

$\Phi_t(.)$        Flow map of the Unknown dynamical system $f(.)$

$f_\theta(.)$        Dynamical system governing the augmented state **u**

$\Phi_{t,\theta}(.)$   Flow map of the dynamical system $f_\theta(.)$

## D  DIMENSIONALITY ANALYSIS OF THE NBEDDYN MODEL

One of the Key parameters of the proposed model is the dimension of the latent space. Despite the fact that it is extremely challenging to get a prior idea of the dimension of the model in the case of real data experiments, one can analyze the spawned manifold of the learnt latent states to get an idea of the true dimension of the underlying model (true here stands for a sufficient dimension of the latent space). The idea here is to compute the modulus of the eigenvalues of the Jacobian matrix for each input of the training data. An eigenvalue does not influence the temporal evolution of the latent state if it has a modulus that tend to zero. The number of non-zero eigenvalues can then be seen as a sufficient dimension of the latent space.

Regarding the identification of an ODE model governing the first state variable of the Lorenz 63 model, Fig. 3 illustrates the eigenvalues of the Jacobian matrix and their modulus for a dimension of the latent space $d_E = 6$. Interestingly, only 3 eigenvalues have non-zero modulus and are effectively influencing the underlying dynamics. This result shows that one can use a 3 dimensional latent-space as a sufficient dimension to identify an ODE model governing the first state of the Lorenz 63 system which is the same dimension as the true Lorenz 63 model.

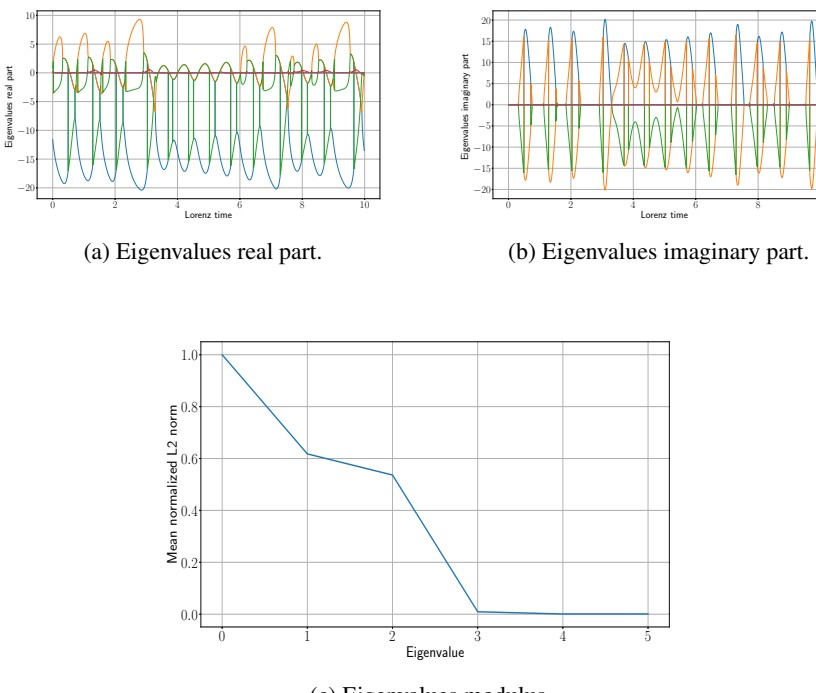

(a) Eigenvalues real part.

(b) Eigenvalues imaginary part.

(c) Eigenvalues modulus.

Figure 3: *Analysis of the eigenvalues of the NbedDyn model Jacobian matrix.*: Lorenz-63 case-study with $d_E = 6$

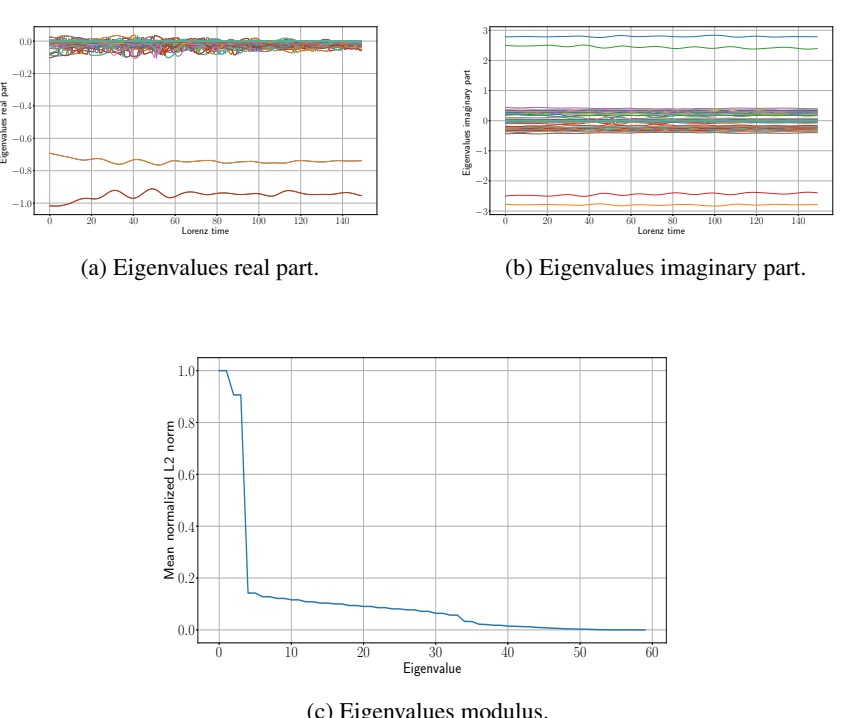

(a) Eigenvalues real part.

(b) Eigenvalues imaginary part.

(c) Eigenvalues modulus.

Figure 4: *Analysis of the eigenvalues of the NbedDyn model Jacobian matrix.*: Sea Level Anomaly case-study with $d_E = 60$

The analysis of the eigenvalues of the Sea Level Anomaly model in the other hand are not as straight-forward as in the case of the Lorenz model since we do not have any idea on the analytical form of the underlying dynamical model. Fig. 4 illustrates that using a 60 dimensional latent space for the NbedDyn model, only 50 eigenvalues have non-zero modulus and thus, are effectively influencing the underlying dynamics. The conclusion in this case is that the observed SLA data evolve in a 50 dimensional latent space parametrised by the dynamical model $f_\theta$.

# E    ADDITIONAL FIGURES OF THE LORENZ 63 EXPERIMENT

We illustrate the forecasting performance of the tested models for the Lorenz-63 experiment through an example of forecasted trajectories in Fig. 5. Our model with $d_E = 6$ leads to a trajectory similar to the true one up to 7 Lyapunov times, when the best alternative approach diverge from the true trajectory beyond 4 Lyapunov times.

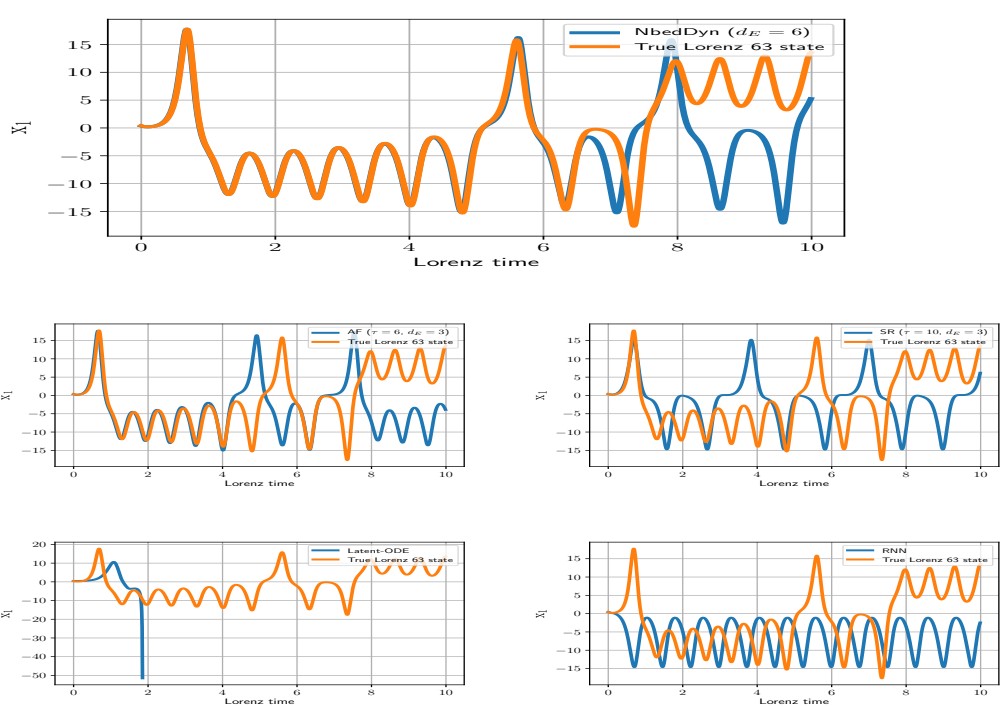

Figure 5: *Generated time series of the proposed models.* : Given the same initial condition, we generated a time series of 1000 time steps.

An other interesting experiment is to find the initial condition for new observation data. This issue is addressed as presented in section 3 as follow. Given a new noisy and partial observation sequence (Fig. 6), we first look for a potential initial condition in the inferred training latent state sequence. This initial condition is then optimized using the cost function described by equation (7) to minimize the forecasting error of the new observation sequence.

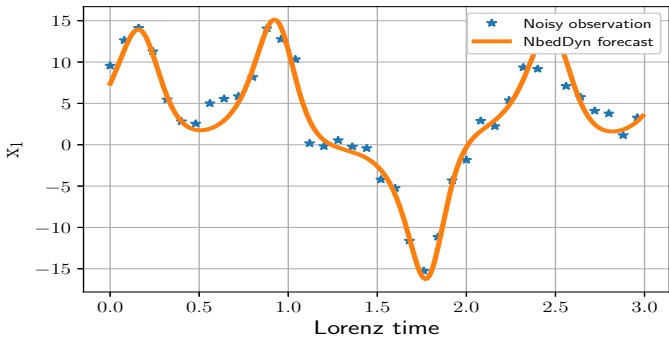

Figure 6: **Forecasted Lorenz 63 state sequence given noisy and partial observations**: Given noisy and partial observations, our model optimizes equation (7) to infer an initial condition that minimize the forecasting of the observations.

## F  ADDITIONAL FIGURES OF THE SEA LEVEL ANOMALY EXPERIMENT

Forecasted states of the Sea Level Anomaly are illustrated in Fig. 8 and 9. The visual analysis of the forecasted SLA states emphasize the relevance of the proposed NbedDyn model. While state of the art approaches generally overestimate the time evolution of some structures such as eddies, our model is the only one to give near perfect forecasting up to 4 days.

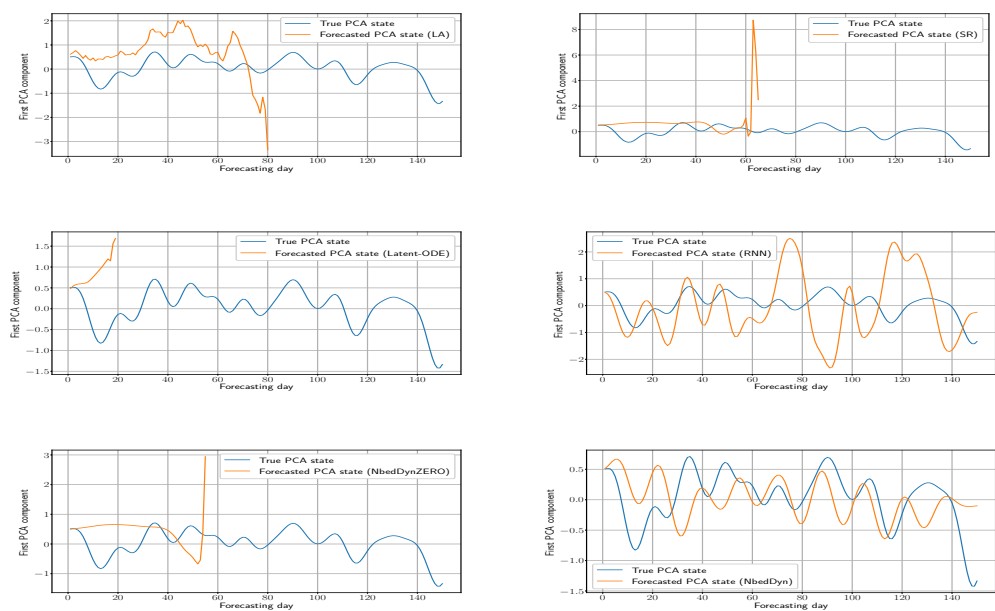

Figure 7: **Generated time series of the proposed models for the forecasting of the SLA dynamics.** : Given the same initial condition, we generated a time series of 150 days.

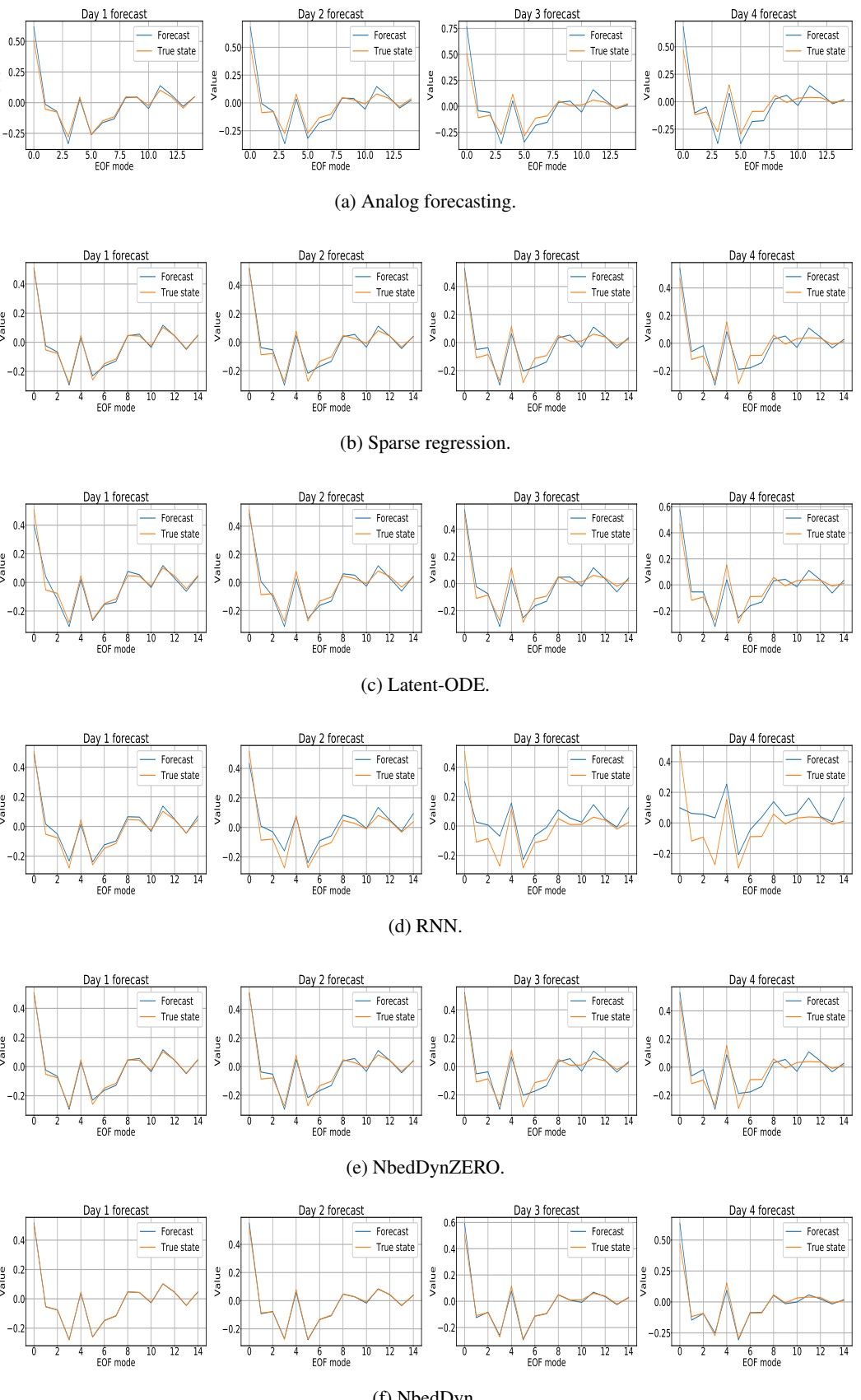

(a) Analog forecasting.

(b) Sparse regression.

(c) Latent-ODE.

(d) RNN.

(e) NbedDynZERO.

(f) NbedDyn.

Figure 8: *Forecasted EOF components of the proposed models.*

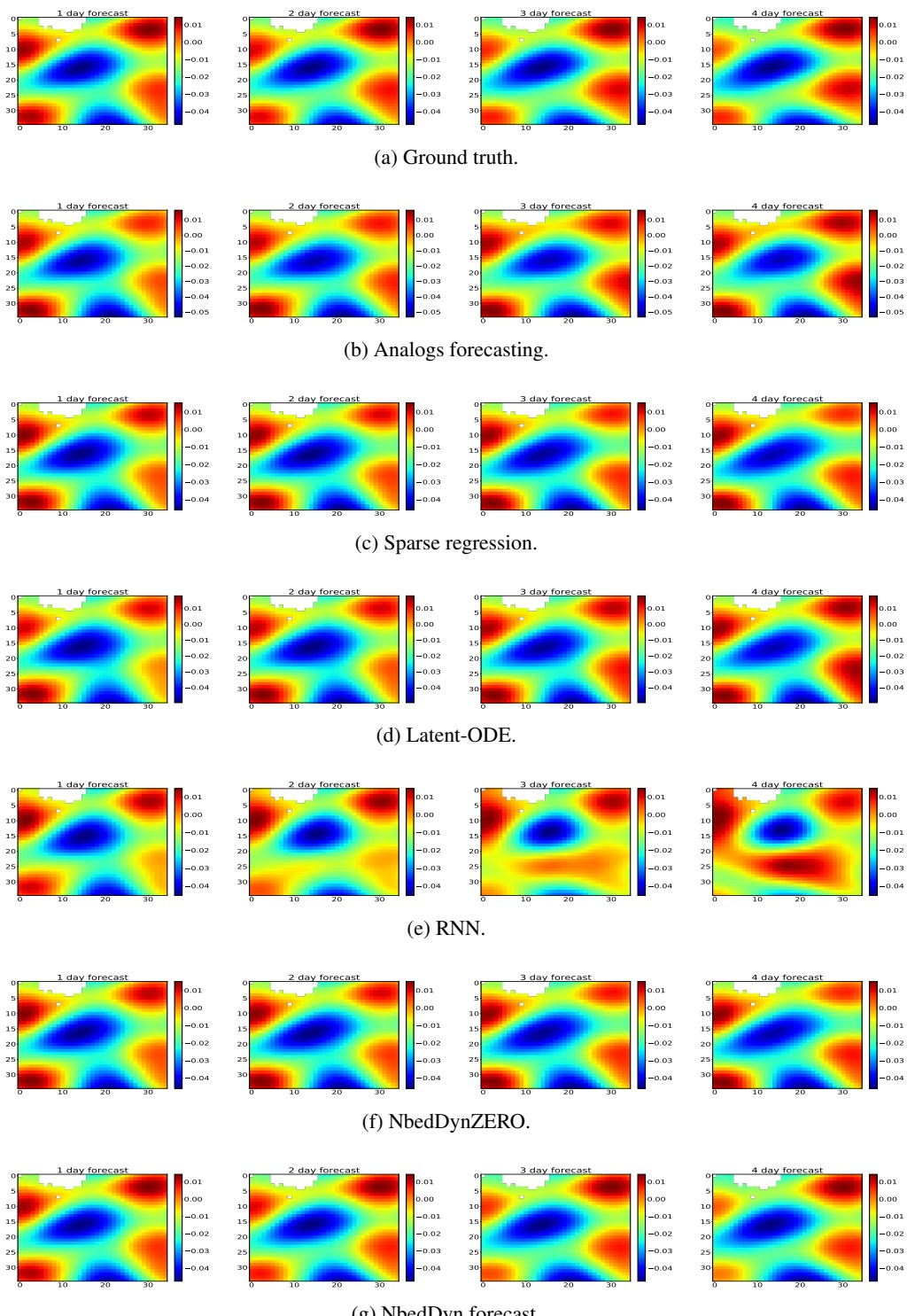

Figure 9: *Forecasted SLA states of the proposed models.*

# G  NEURAL NETWORKS HYPERPARAMETERS

The neural-network parametrization for the dynamical model operator $f_\theta$ is a bilinear architecture as presented in (Fablet et al., 2018). The idea of this model is to embed second order polynomial repre-

sentations of the input states by combining fully-connected layers and an element-wise product. This bilinear layer are then concatenated to classical fully connected layers to perform regression. This architecture incorporates physical knowledge on the dynamics since physical dynamical systems often involve bilinear non-linearities. Among classic physical models, one may cite for instance advection-diffusion dynamics or shallow water equation.

This model is than integrated numerically to compute the flow-map $\Phi_{\theta,t}$ using a fourth order Runge Kutta scheme. In practice, the integration scheme is implemented as a ResNet with four layers sharing the dynamical model.

## G.1 LORENZ 63 EXPERIMENTS HYPERPARAMETERS

| Parameter | Value |
|---|---|
| Number of LSTM layers | 10 |
| Hidden size | 10 |
| Sequence length | 30 |
| Learning rate | 0.001 |
| Optimizer | Adam |
| Training data | 4000 |

Table 3: RNN parameters in the Lorenz 63 Experiment.

| Parameter | Value |
|---|---|
| Latent dimension | 4 |
| Hidden size | 15 |
| RNN hidden size | 100 |
| Learning rate | 0.01 |
| Optimizer | Adam |
| Training data | 4000 |

Table 4: Latent-ODE parameters in the Lorenz 63 Experiment, please refer to (Chen et al., 2018) for more details.

| Parameter | Value |
|---|---|
| Augmented Latent dimension | 6 |
| Number of bilinear layers | 6 |
| Number of linear layers | 6 |
| Integration scheme | Runge Kutta 4 |
| Learning rate | 0.001 |
| Optimizer | Adam |
| Training data | 4000 |

Table 5: NbedDyn parameters in the Lorenz 63 Experiment, please refer to (Fablet et al., 2018) for more details.

## G.2 SLA EXPERIMENTS HYPERPARAMETERS

| Parameter | Value |
|---|---|
| Number of LSTM layers | 5 |
| Hidden size | 20 |
| Sequence length | 40 |
| Learning rate | 0.001 |
| Optimizer | Adam |
| Training data | 2000 |

Table 6: RNN parameters in the SLA Experiment.

| Parameter | Value |
|---|---|
| Latent dimension | 60 |
| Hidden size | 70 |
| RNN hidden size | 200 |
| Learning rate | 0.01 |
| Optimizer | Adam |
| Training data | 2000 |

Table 7: Latent-ODE parameters in the SLA Experiment, please refer to (Chen et al., 2018) for more details.

| Parameter | Value |
|---|---|
| Augmented Latent dimension | 60 |
| Number of bilinear layers | 60 |
| Number of linear layers | 60 |
| Integration scheme | Runge Kutta 4 |
| Learning rate | 0.001 |
| Optimizer | Adam |
| Training data | 2000 |

Table 8: NbedDyn parameters in the SLA Experiment, please refer to (Fablet et al., 2018) for more details.

## H    SCOOP AND LIMITATIONS

**Constraining limit cycles** The proposed augmented ODE formulation does not suppose any prior knowledge on the underlying dynamics responsible for the temporal evolution of the observations. This can lead in some cases (especially when working on chaotic dynamics) to output a dynamical representation that has several attracting regions in addition to the one leading to the observations limit cycle. This can lead to inappropriate results when trying to find an initial condition that forecasts a given observation sequence. The Idea of using the manifold spanned by the augmented training data allows to bypass this issue but we believe that adding additional constraints (energy preserving constraints, known symmetries in the models ...etc.) can significantly improve the quality of the learnt dynamical models.

## I  CODE SAMPLE

```
# import libs
from generate_data import generate_data
import numpy as np
import torch
from torch.autograd import Variable
seed = 0
np.random.seed(seed)
torch.manual_seed(seed)
#Generate data
class GD:
    model = 'Lorenz_63'
    class parameters:
        sigma = 10.0
        rho = 28.0
        beta = 8.0/3
    dt_integration = 0.01 # integration time
    nb_loop_data = 50.01 # size of the catalog
    test_samples = 1000
# run the data generation
catalog, xt, yo = generate_data(GD)
X_test    = catalog.analogs[catalog.analogs.shape[0]-GD.test_samples:,:1]
X_train   = catalog.analogs[:catalog.analogs.shape[0]-GD.test_samples,:1]
Grad_t    = np.gradient(X_train[:,0]).reshape(X_train.shape[0],1)\
/GD.dt_integration
x = Variable(torch.from_numpy(X_train).float())
z = Variable(torch.from_numpy(Grad_t).float())
data_size = X_train.shape[0]
#neural net params
params = {}
params['transition_layers']=1
params['bi_linear_layers']=6
params['dim_hidden_linear'] = 6
params['dim_input']=1
params['dim_latent']=5
params['dim_output']=params['dim_input'] + params['dim_latent']
params['dt_integration'] = GD.dt_integration
#Dynamical model
class FC_net(torch.nn.Module):
        def __init__(self, params):
            super(FC_net, self).__init__()
            y_aug = np.random.uniform(size=(data_size, params['dim_latent']))
            self.y_aug = torch.nn.Parameter(torch.from_numpy(y_aug).float())
            self.linearCell   = torch.nn.Linear(params['dim_output']\
                                                , params['dim_hidden_linear'])
            self.BlinearCell1 = \
            torch.nn.ModuleList(\
            [torch.nn.Linear(params['dim_output'], 1)\
            for i in range(params['bi_linear_layers'])])
            self.BlinearCell2 = \
            torch.nn.ModuleList(\
            [torch.nn.Linear(params['dim_output'], 1)\
            for i in range(params['bi_linear_layers'])])
            augmented_size    = params['bi_linear_layers']\
            + params['dim_hidden_linear']
            self.transLayers = \
            torch.nn.ModuleList(\
```

```
                    [torch.nn.Linear(augmented_size, params['dim_output'])])
                    self.transLayers.extend(\
                    [torch.nn.Linear(params['dim_output'], params['dim_output'])\
                    for i in range(1, params['transition_layers'])])
                    self.outputLayer = torch.nn.Linear(params['dim_output'],\
                                                        params['dim_output'])
            def forward(self, inp, dt):
                if inp.shape[-1]<params['dim_latent']+params['dim_input']:
                    aug_inp = torch.cat((inp, self.y_aug), dim=1)
                else:
                    aug_inp = inp
                BP_outp = Variable(torch.zeros((aug_inp.size()[0],\
                                                 params['bi_linear_layers'])))
                L_outp    = self.linearCell(aug_inp)
                for i in range((params['bi_linear_layers'])):
                    BP_outp[:,i]=self.BlinearCell1[i](aug_inp)[:,0]*\
                    self.BlinearCell2[i](aug_inp)[:,0]
                aug_vect = torch.cat((L_outp, BP_outp), dim=1)
                for i in range((params['transition_layers'])):
                    aug_vect = (self.transLayers[i](aug_vect))
                grad = self.outputLayer(aug_vect)
                return grad, aug_inp
model  = FC_net(params)
# compute flow : RK4
class INT_net(torch.nn.Module):
        def __init__(self, params):
            super(INT_net, self).__init__()
#              self.add_module('Dyn_net',FC_net(params))
            self.Dyn_net = model
        def forward(self, inp, dt):
            k1, aug_inp   = self.Dyn_net(inp,dt)
            inp_k2 = aug_inp + 0.5*params['dt_integration']*k1
            k2, tmp   = self.Dyn_net(inp_k2,dt)
            inp_k3 = aug_inp + 0.5*params['dt_integration']*k2
            k3, tmp   = self.Dyn_net(inp_k3,dt)
            inp_k4 = aug_inp + params['dt_integration']*k3
            k4, tmp   = self.Dyn_net(inp_k4,dt)
            pred = aug_inp +dt*(k1+2*k2+2*k3+k4)/6
            return pred, k1, inp, aug_inp
#Instanciate the model
modelRINN = INT_net(params)
criterion = torch.nn.MSELoss(reduction='elementwise_mean')
optimizer = torch.optim.Adam(model.parameters())
#Pretraining : fit the gradient
params['ntrain']=[300000,10000]
for t in range(params['ntrain'][0]):
        pred, grad, inp, aug_inp = modelRINN(x,params['dt_integration'])
        loss1 = criterion(grad[:,:1], z)
        loss2 = criterion(pred[:-1,:], aug_inp[1:,:])
        loss = 0.9*loss1+0.1*loss2
        print(t,loss)
        optimizer.zero_grad()
        loss.backward(retain_graph=True)
        optimizer.step()
# training
for t in range(params['ntrain'][1]):
        pred, grad, inp, aug_inp = modelRINN(x,params['dt_integration'])
        loss1 = criterion(pred[:-1,:], aug_inp[1:,:])
        loss2 = criterion(pred[:-1,1:], aug_inp[1:,1:])
```

```
loss  =10.0* loss1  +  1.0* loss2
print ( t , loss )
optimizer . zero_grad ()
loss . backward ( retain_graph =True )
optimizer . step ()
```

