# OpenReview forum: "Learning Latent Dynamics for Partially-Observed Chaotic Systems"
_ICLR.cc/2020/Conference — Reject_

### Official Review · AnonReviewer2 · 2019-10-22
**Official Blind Review #2**

**Rating:** 3

**Review:**

The paper proposes a new deep learning approach based on Takens’s theorem to identify the dynamics of partially observed chaotic systems. In particular, the method augments the state using the solution of an ODE. Experiments on Lorenze-64 dynamics and sea level anomaly demonstrate the advantage of the proposed method over state-of-the-art baselines.

+ The unification of Taken’s embedding theorem and deep learning provides a novel perspective into dynamical systems
+ Impressive experiment results compared with baselines including RNN and latent ODE

- The proposed method requires knowledge of the underlying dynamic model to solve the ODE, which is not fair for other methods
- The model is trained using data from the same initial conditions, which is essentially overfitting. The authors should provide experiments for dataset from different initial conditions.
- The writing is not very clear. For example, how to solve the optimization problem in Eqn (7),  as the augmented states u_{t-1} are unknown? How to find the bijective mapping M for general dynamical systems?

Minor: question mark in section 4 page 6.   Figures 2 plots are difficult to read, pls provide more details in columns and rows.

**Experience Assessment:**

I have published one or two papers in this area.

**Review Assessment: Checking Correctness Of Derivations And Theory:**

I assessed the sensibility of the derivations and theory.

**Review Assessment: Checking Correctness Of Experiments:**

I assessed the sensibility of the experiments.

**Review Assessment: Thoroughness In Paper Reading:**

I read the paper at least twice and used my best judgement in assessing the paper.

---

> ### Author Response · Authors · 2019-11-13
> **Reviewer 2 answers**
>
> The authors would like to thank the reviewer for his valuable comments and suggestions, please find below the our answers.
>
> >> "The proposed method requires knowledge of the underlying dynamic model to solve the ODE, which is not fair for other methods."
>
> a : The proposed model does not requires more knowledge of the underlying dynamic model to solve the ODE than the benchmarked models. As stated in page 4 (page 5 in the revised manuscript), learning scheme section, both the latent states and the parameters of the approximate model (which is parametrized as a neural network with the parametrization given in the appendices) are learnt from the set of observations. For instance, the Neural ODE baseline (Latent-ODE) uses the exact same dynamical model as the proposed framework.
>
> >> "The model is trained using data from the same initial conditions, which is essentially overfitting. The authors should provide experiments for dataset from different initial conditions."
>
> a : The model is not trained using the same initial condition. As stated in equation (6), the learning criterion is simply a one step ahead forecast (as used in most of the regression issues). All the points of the observed time series are used as initial conditions of the model and the one step ahead forecast is matched using MSE to the true next state.
>
> >> "The writing is not very clear. For example, how to solve the optimization problem in Eqn (7), as the augmented states u{t-1} are unknown? How to find the bijective mapping M for general dynamical systems?."
>
> a : We apologize for the confusion that might have been caused by our writing. Several paragraphs and sentences were updated in each section, please refer to the revised version of the manuscript for more details. The optimization problem is solved simply by considering the latent variables of u as parameters of the loss function, so we can use automatic differentiation to compute the gradients of the loss function with respect to the model parameters and with respect to the latent states. We then update both the model parameters and the latent states using classical gradient decent techniques.
>
> The derivation of the bijective mapping M depends on the application. For example, regarding the derivation of reduced order models to spatio-temporal fields, Galerkine projections of fluid flows are usually considered since it comes with some physical interpretability of the projection. Here, for the SLA case-study, M is just a PCA projection. In more complex situations such as in [1], one can use an autoencoder to actually learn the mapping M. This can be done offline or online with the learning of the dynamical model parameters and the latent states. The latter technique is particularly interesting and is considered as one of our future works.
>
> For the sake of simplicity we ommited the use of the operator M in the mathematical developpements. we added a section in the appendix to derive the same equations in the case of reduced order models.
>
> [1] Champion, Kathleen, et al. "Data-driven discovery of coordinates and governing equations." arXiv preprint arXiv:1904.02107 (2019).
>
> Finally the minor comments were addressed in the revised manuscript.

---

### Official Review · AnonReviewer3 · 2019-10-22
**Official Blind Review #3**

**Rating:** 3

**Review:**

Update: I raised the score from 1 to 3 to acknowledge the authors' consideration for the 2000-2010 literature on learning dynamical systems from partial observations. Unfortunately, the writing is still confusing, some of the claims in the introduction and rebuttal are inexact ([5] does not embed the observations and does work with partially observed environments), and the method lacks originality compared to existing work. Other work that relies on ODE integration and in finding high-dimensional state variables has recently been published and tested on more ambitious datasets, e.g. Ayed et al, "Learning Dynamical Systems from Partial Observations", arXiv 2019.

***

TL;DR: relatively well written (if sometimes confusing) paper that reinvents the inference of latent variables in nonlinear dynamical systems that has been published in the 2000s, and that misses an important chunk of literature (and experiments on dynamical systems such as Lorenz-63) from that time.

This paper proposes an approach for learning dynamical systems from partial observations x, by using an augmented state variable z that follows dynamics that can be described by an ordinary differential equation (ODE) with dynamics f. The authors motivate their work by the problem of dynamical system identification when only partial observations are available. The authors claim that was to date primarily addressed using time-delay embedding, following Takens' theorem. The authors introduce s-dimensional unknown state variables z, dynamical function f (for the ODE on z), flow phi on z, limit cycles on z, observation function H: z -> x that goes from z to n-dimensional observations x, low k-dimensional manifold r (with a map M: x -> r), and state augmentation variable y. The reconstructed state variable u is the concatenation of r and y. One key ingredient of the method is to infer the optimal value of state augmentation variable y during learning (see equations 5 and 6) and inference for forecasting (7); this is not well explained in the abstract and introduction.

I would note that the problem of state space modeling (SSM) and dynamical system identification has been well studied, and the notation and reformulation in this paper is somewhat confusing for those who are used to the notation in SSMs (specifically, expressing the observation approximation as M^{-1}(G(phi(u_{t-1}))). Learning a state-space model involves both learning parameters and inferring the latent states representation (or, in graphical models, the distribution of these latent states) given the parametric models. One approach has been to formulate the state-space model learning by maximum likelihood learning of the model parameters so that the generative model fits observed data x, and this would involve factoring out the distribution of latent states z; the algorithm would rely on Expectation Maximisation, and could involve variational approximations or sampling. While the state space models were hampered by their linearity, several papers in 2000s showed how it is possible to learn nonlinear dynamical models, e.g. [4], [5], [6] and [7] to cite earlier ones. Equations (5) and (6) are similar to the standard equations for a dynamical system expressed in continuous time, with the only difference that the optimisation is with respect to y only, rather than w.r.t. z or u (why not \tilde z or \hat z?).

The paper mentions various initialisation strategies for y (last paragraph of section 3). Why not predict from the past of the observations, like is done in many other similar work?

The literature review mixes older and newer references. For example, on page 1, I would note that the Takens' theorem has been applied in conjuction with Support Vector Regression as early as 1999 [1][2], and with neural networks in 1993 [3].

Most importantly, the ideas of this paper have already been published in [4] (with architecture constraints on the neural network state-space model), in [5] (with any nonlinear neural network state-space model), in [6] (using Restricted Boltzmann Machines) and in [7] (using Gaussian Process latent variable models).
The model is illustrated with experiments on a 2D linear attractor, on the Lorenz-63 attractor. Given the results published in [1] and [2] using SVR on 1D observations of that attractor, and in [5] using a recurrent neural network, I am unconvinced by these results. It seems in particular that the number of training points (around 4000) limits the performance of RNN / LSTM models. The application to Sea Level Anomaly is interesting.

Minor comments:
"Unfortunately, When" (page 1)
There is a missing -1 after M in equation (5) and (10)
In equation (7), should not the sum go from t=0 to T, as x_t is unknown for t>T?
What is prediction and what is extrapolation on Figure 1?
The caption of Fig 1 contains (left)
The figures seem squeezed with the captions / titles un-aesthetically wide.
Labels on Figure 5 in the appendix seem mixed, and red should be the ground truth

[1] Mattera & Haykin (1999) "Support vector machines for dynamic reconstruction of a chaotic system"
[2] Muller, Smola, Ratsch, Scholkopf, Kohlmorgen & Vapnik (1999) "Using support vector machines for time-series prediction"
[3] Wan (1994) "Time series prediction by using a connectionist network with internal delay lines"
[4] Ghahramani, and Roweis (1999) "Learning nonlinear dynamical systems using an EM algorithm"
[5] Mirowski & LeCun (2009) "Dynamic Factor Graphs for Time Series Modeling"
[6] Taylor, Hinton & Roweis (2006) "Modeling human motion using binary latent variables"
[7] Wang, Fleet & Hertzmann (2006) "Gaussian process dynamical models"


**Experience Assessment:**

I have published one or two papers in this area.

**Review Assessment: Checking Correctness Of Derivations And Theory:**

I carefully checked the derivations and theory.

**Review Assessment: Checking Correctness Of Experiments:**

I carefully checked the experiments.

**Review Assessment: Thoroughness In Paper Reading:**

I read the paper thoroughly.

---

> ### Author Response · Authors · 2019-11-13
> **Reviewer 3 answers 3**
>
> Regarding the definition of the experimental setup, we considered a training dataset with 4000 points, which is a trade-off w.r.t. previous works which performed experiments with 2000 [5] and 10000 [1] points. To our knowledge, none of these previous works compute Lyapunov exponents to evaluate the long-term behaviour of the learnt models, which make direct comparisons with published results more complex. This is the reason why we report experiments with different models compared within the same experimental setup. Benchmarked models include both "old" state-of-the-art schemes that do achieve reasonably good long-term patterns on Lorenz-63 dynamics, e.g. analog schemes combined with a delay embedding, state-of-the-art ODE inference schemes combined with delay embedding schemes (Sparse regression method and Latent-ODE scheme) and a simple RNN model. We may also point out that Latent-ODE exploits a state-of-the-art deep learning framework (pytorch) and involves a LSTM mapping to infer the latent state from the observation series. As such, it may also be regarded as being representative of LSTM-based deep learning models using dely embeddings with the additional benefit of identifying an ODE representation as targeted in our work.
>
> Finally, all the minor comments were addressed in the revised version of the manuscript. Specifically, the operator M was omitted in the mathematical development of the paper to provide a more friendly written development similar to classical SSM. The generalization of the approach to ROM (including the operator M) is now in the appendix.
>
> [1] Mattera & Haykin (1999) "Support vector machines for dynamic reconstruction of a chaotic system"
> [2] Muller, Smola, Ratsch, Scholkopf, Kohlmorgen & Vapnik (1999) "Using support vector machines for time-series prediction"
> [3] Wan (1994) "Time series prediction by using a connectionist network with internal delay lines"
> [4] Ghahramani, and Roweis (1999) "Learning nonlinear dynamical systems using an EM algorithm"
> [5] Mirowski & LeCun (2009) "Dynamic Factor Graphs for Time Series Modeling"
> [6] Taylor, Hinton & Roweis (2006) "Modeling human motion using binary latent variables"
> [7] Wang, Fleet & Hertzmann (2006) "Gaussian process dynamical models"
> [8] Krishnan, Rahul G., Uri Shalit, and David Sontag. "Deep Kalman Filters.(2015)." arXiv preprint arXiv:1511.05121 (2015).
> [9]Fraccaro, Marco, et al. "Sequential neural models with stochastic layers." Advances in neural information processing systems. 2016.
> [10] Chen, Tian Qi, et al. "Neural ordinary differential equations." Advances in neural information processing systems. 2018.

---

> > ### Comment · AnonReviewer3 · 2019-11-15
> > **Official Blind Review #3 update**
> >
> > Thank you for the long and detailed review and rebuttal. I am now reading the updated version of the paper.

---

> ### Author Response · Authors · 2019-11-13
> **Reviewer 3 answers 2**
>
> >> "The literature review mixes older and newer references ..."
>
> a : We Thank the reviewer for these suggestions. We added the suggested references in the introduction.
>
> >> "Most importantly, the ideas of this paper have already been published..."
>
> a : We would like to thank the reviewer for the good feedback on the SLA application. We argue that the proposed methodology is significantly different from the previous work pointed out by the reviewer. In addition to the fact that none of these interesting papers addresses the identification of an ODE representation for partially-observed systems, we may stress the following points (paper numbers corresponding to the reviwers comment not to our papers numbers):
>
> Paper [1]: This paper also addresses the long-term simulation of the trajectories of the system using a time delay embedding and SVR in the defined latent space. As mentioned above, a critical aspect of such schemes is the selection of the time delay. In [1], reported results show that simulated trajectories match real ones up to 4 Lorenz time using 10000 training data. We reach a similar performance using 4000 training data. In our experiments, we used as baselines time delay embeddings combined with nearest-neighbor (analog methods) and sparse regression (SR) techniques. The later is very similar to a SVM. It actually relates to an ODE formulation and accounts for non-linear (polynomial) terms in the ODE, which is known to be critical to reproduce chaotic Lorenz-63 dynamics. Our experiments point out that we reach much better performance than those baselines both in terms of forecasting performance and long-term behaviour (cf. largest Lyapunov exponents in Tab.1). The reported results also stress that the inference of the largest Lyapunov exponent, i.e. the ability of the learnt model to reproduce realistic long-term patterns, is very sensitive to the considered delay embeddings.
>
> Paper [4]: this paper does not consider the case of partially observed systems since in the presented experiment the latent states can be fully determined given the observations, which is a much simpler experimental setup. As such, it does not involve the definition of some augmented state. From a methodological point of view, the authors use an  Ensemble Kalman filtering scheme to reconstruct the latent state from the observation series. Here, we prefer considering a variational setting, which is similar to 4D-Var assimilation scheme. This choice appears more natural to provide an end-to-end learning framework using deep learning framework.
>
> Paper [5] : From a modeling perspective, this paper states the dynamics of the latent states as a function of the previous latent states as well as of the previous observations. As such, it relies on an explicit Takens's delay embedding strategy, for which the selection of the time delay is critical. In our work, there is no such requirement for selecting a time delay. Importantly, [5] does not aim to identify an ODE representation. Besides, to the best of our knowledge, the suggested paper [5] does not achieve  state-of-the-art (based on delay embedding) attractor reconstruction using closed loop forecast as the presented attractor reconstruction application in [5] is just the projection of the observed variables in the latent space. Similar results can be obtained using an appropriate delay embedding without any dynamical model.
>
> Paper [6] : The proposed architecture models the dynamics in the observation space and uses a binary latent space as a proxy for increasing the expressiveness of the model. Although this technique seems relevant for modeling periodic human motion, a huge limitation of this model is that it cannot model chaotic behavior (as discussed in the paper since two different regimes as running and walking can not be generated from the same initial condition).
>
> Paper [7] : The proposed model is limited in terms of architecture since it uses RBF approximations for the dynamical and observation models. The proposed architecture aims to infer low-dimensional latent space where the dynamics of the observations evolve (which is represented in our paper by the operator M). This architecture is similar to the tested Latent-ODE model where the training of the model is based on likelihood maximization of the posterior over the latent states given the observations. As stated above, in our scheme, we do not explicitly constrain the mapping between the observation series and latent space according to some non-linear (possibly complex) model. We only consider an implicit mapping through a minimization issue. Furthermore, in [7], the results are presented only for a short term forecast application (8 frames) where in our paper we are also interested in the long forecast of the model.

---

> ### Author Response · Authors · 2019-11-13
> **Reviewer 3 answers 1**
>
> The authors would like to thank the reviewer for his valuable comments and suggestions, please find below the our answers.
>
> >> "Relatively well written (if sometimes confusing) paper that reinvents the inference of latent variables ..."
>
> a : We apologize for the confusion that might have been caused by our writing. We thank the reviewer for pointing out interesting related papers. We however believe that our formulation differs from those previous works as clarified below.
>
> >> "This paper proposes an approach for learning dynamical systems from .."
>
> a : We wanted to give an overall picture of the issue in the introduction and related works, that is why the equations and notations raised by the reviewer only appear in the 3rd section. We apologize for the confusion.
>
> >> "I would note that the problem of state space modeling (SSM).."
>
> a : We agree that SSM are extremely well studied in the literature. The key difference with the proposed approach is that, to our knowledge, advances in latent inference in state space models was introduced essentially, from a dynamical systems perspective, to find low dimensional manifolds, where the dynamics of the system evolve (which is considered in our paper and the literature review on ROM through the mapping M). When used in the context of partially-observed systems, the latent variables are inferred from a sequence of observations (which is pretty much similar to a parametric delay embedding) through parametric modeling of the posterior distribution as in [8,9,10] or through marginalization with models constraints as in [4,7]. They however often fail to model long term patterns using the learnt dynamical model (as shown in the experiments). This is due to the fact that the latent space is constrained to be a non linear projection of a sequence of observations, which we believe limits the expressiveness of the dynamical model. The other solutions explored in the literature usually exploit  delay embeddings of the observations. For instance, in [6], the dynamics of the latent states depend on both the previous latent states as well as a delay embedding of the observations. By contrast, we aim to identify an autonomous ODE representation in the latent space, i.e. an ODE which drives the dynamics of the latent state with observation-related forcing.
>
> Overall, when considering an ODE representation in the latent space, we may regard these different approaches as different ways to solve for minimization (6). The key finding of our paper is to show that (i) when considering data driven derivation of ODE representations of some observed data, one should make sure that the observations form an embedding of the true underlying system, otherwise the ODE may not be able to simulate the observations (ii) when considering an approximate augmented latent space for modeling partially observed systems, there is no need to introduce an inference model to find the latent states of the dynamics as this can limit the expressiveness of our model, (iii) the latent states can be derived from a numerical optimization problem that can be solved using the numerical optimization tools of deep learning packages with no additional constraint, (iiii) these differences significantly impact both shot-term forecast performance and, more importantly, the ability to recover the long-term chaotic behaviour.
>
> Latent-ODE [7] is regarded as a state-of-the-art ODE approach where one jointly learns the ODE in the latent space and a LSTM-based inference model for the latent states given the observed time series. We believe that this technique is representative of inference in state space models and shows that if the latent states are computed as a non linear combination of an observation series, the forecasting performances of the model is significantly reduced.
>
> >> "The paper mentions various initialisation strategies for y ..."
>
> a : For an application to forecasting, we need to infer the initial condition in the latent space (more precisely, the unknown component y of the augmented state) from a given past observation series. This is stated as a minimisation issue (Eq.7), i.e. the inference of the latent state sequence which best match the past observation series w.r.t. a learnt ODE model. Similarly to the training phase, this inference is solved using a gradient descent. As in any gradient descent algorithm, one has to set an initialization. Different initialization strategies were explored, especially strategies which benefit from the latent state sequences inferred during the training phase.
>
> In most related work, as mentioned above, no such minimization is required as one defines an explicit mapping between the observed time series and the latent state sequence. For instance, in Latent-ODE scheme, one would first predict the initial latent state from a past observation series using a learnt mapping model. Second, an ODE integration scheme is applied from this initial condition to predict future states.

---

### Official Review · AnonReviewer1 · 2019-10-31
**Official Blind Review #1**

**Rating:** 6

**Review:**

The paper addresses the problem of data-driven identification of latent representations for partially-observed dynamical systems. The proposed method uses an augmented state-space model where the dynamical operator is parameterized using neural networks. Given some training data, the proposed approach jointly estimates the parameters of the model as well as the corresponding unobserved components of the latent space for the training sequences. Experimental results on forecasting using real and synthetic datasets are presented.

Overall I think that the paper makes an interesting contribution. I find very interesting the idea of performing inference on the set of unobserved components in the latent space. The empirical results seem sufficient to me, but I am not familiar with relevant baselines (see below). Please find below some comments and questions.

I am personally not familiar with the literature on this problem, so my assessment might be affected by this. I did not find the paper easy to read and the presentation assumes a lot of previous knowledge. I think that the background and related work section could be more friendly written (considering the ICLR audience).

The training scheme (described in Section 3) uses one-step-ahead forecasting. The temporal consistency of the unobserved component of the latent space is only loosely enforced with the regularization term in (6). One could train using forecasting with more steps (and only doing inference for the initial y_t of the subsequence), as this is closer to what is used at test time. Do you think this would be helpful for having better accuracy when forecasting more steps?

It would be good to provide more details on how to build the forecasting operator (implementing 4-th order Runge-Kutta scheme) and what is exactly the bilinear architecture of Fabelt et al.

Regarding the experimental validation, I like that the paper starts with a simple motivating example and moves to more complex cases. Experimental results are convincing to me, as the model is able to recover the performance of other models that do have access to the full state. I am not familiar with the literature so I'm unable to judge whether all relevant baselines are included.

Regarding the Latent-ODE baseline, would results change running with different (larger) dimension for the latent space?

The paper should cite the work: Ayed, et al. "Learning Dynamical Systems from Partial Observations." arXiv preprint arXiv:1902.11136 (2019). Would this be a relevant baseline to compare to?

Is the training data regularly-sampled? Would the model be robust the irregularly-sampled training data?

The authors evaluate all methods with one and four step forecasting in the last two experiments. I think that it would be informative to show a wider range of number of steps, to show how performance degrades with longer predictions (more than 4).

Finally, regarding the Modelling Sea Level Anomaly task, all baselines are ran by the authors. It would be informative to also include results of prior art using this dataset, if possible.

Other minor comments:

The citation format is wrong. Most citations should be using the \citep command

In the second paragraph of Section 1, it says: "Unfortunately, When the"

In the caption of figure 2 it says: "according to thr"

A few lines before the "Modelling Sea Level Anomaly" subsection there's an exclamation sign before the text"1e-4 for a one-step..."


**Experience Assessment:**

I do not know much about this area.

**Review Assessment: Checking Correctness Of Derivations And Theory:**

I assessed the sensibility of the derivations and theory.

**Review Assessment: Checking Correctness Of Experiments:**

I assessed the sensibility of the experiments.

**Review Assessment: Thoroughness In Paper Reading:**

I read the paper thoroughly.

---

> ### Author Response · Authors · 2019-11-13
> **Reviewer 1 answers**
>
> The authors would like to thank the reviewer for his valuable comments and suggestions, please find below the our answers.
>
> >> "I am personally not familiar with the literature on this problem ..."
>
> a : We apologize for the confusion that might have been caused by our writing. We added some sentences in the introduction and related works sections that links what we are trying to achieve with classical state of the art inference in state space models, we hope that this will help the readers to quickly catch our contribution.
>
> >> ".The training scheme (described in Section 3) uses one-step-ahead forecasting ..."
>
> a : We completely agree that training with more steps can improve the forecasting (long term) applications. We may however point out that, since we deal with chaotic systems, any small perturbation will make the system diverge from the ground truth trajectory. Hence, learning on long time series with a classic MSE criterion would not help in capturing the long-term patterns. Though using a MSE criterion on a few time-step ahead could improve the reported results, we show in our experiments that the proposed framework can reach very relevant short-term forecasting using a one-step-ahead MSE criteion. We might also point out that the considered formulation implicitly embeds such a multi-step criterion as the training cost minimizes a sum of squared one-step-ahead forecasting errors and a second term which relates to the dynamical prior, i.e. the fact two successive states should relate up to the integration of the learnt ODE.
>
> >> "It would be good to provide more details on how to build the forecasting operator ... "
>
> a : We thank the reviewer for his comment, we added a paragraph in the appendix to explain the form of the approximate dynamical model (which can be arbitrary) and the implementation of the integration scheme.
>
> >> "Regarding the experimental validation ... "
>
> a : We would like to correct one misunderstanding here. All of the tested models have access to the exact same partial observations. Furthermore, we wanted to use recent and old baselines since up to date, there is still (to our knowledge) no new baseline that can forecast long-term trajectories of chaotic systems as good as the Takens embedding based techniques. Other baselines focus on ODE models since the aim of this paper is to generalise ODE representations to partially-observed systems. The Latent-ODE baseline recently presented in [1] shows that computing the latent states as a non linear function of an observed sequence actually restrict the latent space which gives poor forecasting performances, especially in long term. RNNs can be seen as an alternative to finding latent spaces (which includes both the proposed model and Takens embedding based models) by using time sequences as inputs.
>
> >> "Regarding the Latent-ODE baseline... "
>
> a : "We do not think that using a larger dimension for the latent space can improve the Latent-ODE baseline up to generate chaotic trajectories and the reason is that the latent states are computed as a non-linear function of the inputs (thought the variational approximation of the posterior distribution). We suggest that this explicit parametrization of the mapping between the inputs (observed time series) and the latent state sequences may not lead lead to a good approximation of the solution of the underlying ODE. By contrast, in the proposed framework, no such explicit parametrization is considered and we numerically solved for the inference of the latent states as a minimization issue w.r.t. the learnt ODE."
>
> >> "The paper should cite the work: Ayed, et al ..."
>
> a : Though relevant, the paper of (Ayed et al 2019) assumes the knowledge of the initial state of the sequence (here u) to forecast the observation. In this situation, the scheme proposed by (Ayed et al 2019) does not apply.
>
> >> "Is the training data regularly-sampled? ..."
>
> a : The training data is regularly sampled. We have not addressed yet irregularly-sampled data. It should be relatively direct since in an ODE formulation, we only need to do forecast up to the available data points and compute errors on the available data.
>
> >> "The authors evaluate all methods with one ..."
>
> a : Since we are essentially dealing with chaotic systems, we already know that at some point the trajectories will diverge from the ground truth which will make an RMSE score irrelevant. That is the reason why we prefer to give a topological score (largest Lyapunov exponent) and a plot of the forecasted trajectories (in the appendices) instead of an RMSE plot (we can give a graphic of the RMSE if the reviewer still thinks that it is important).
>
> >> "Finally, regarding the Modelling Sea Level ..."
>
> a : Unfortunately, to the best of our knowledge, there is no other work that ran exactly the same data set for forecasting applications.
>
> [1] Chen, Tian Qi, et al. "Neural ordinary differential equations." Advances in neural information processing systems. 2018.

---

> > ### Author Response · Authors · 2019-11-13
> > **Reviewer 1 answers 2**
> >
> > Finally, all the minor comments were addressed in the revised version of the manuscript.

---

### Author Response · Authors · 2019-11-13
**General comments**

The authors would like to thank the anonymous reviewers for their valuable comments and suggestions. In the revised version of the document, we addressed the issues raised as best as possible. Please find below some general comments.

First of all, the aim of this paper is to propose a framework to derive ODE representations for partially-observed systems. This is a generalization of recently proposed ODE representations of time series to partially-observed systems with an emphasis on chaotic system reconstruction through long term forecasting of the learnt ODEs. We rely on the inference of a latent space. This technique is only used as a tool to generalize ODE representations to partially observed systems. As detailed below (in the reviewers answers), previous works investigated the inference for latent spaces dynamical systems but with different objectives.

We believe that modeling an arbitrary (deterministic) time series using ODEs is extremely interesting since it can benefit from a broad literature on differential equations to stabilise models (through stability analysis of ODE), understanding the phenomenons underlying the observations (through PDF analysis of the trajectories using differential transport).... On both a synthetic dataset and a real case-study (sea surface dynamics), we show that the proposed framework can significantly improve short-term forecasting performance and reproduce long-term chaotic behaviour.

We may clarify these two objectives from an experimental point of view. This is actually discussed in one of papers pointed out by reviewer 3 [1]. One may emphasize the difference between short term forecasting applications and dynamical system reconstruction using models which is essentially long term forecasting of the approximate models. The later is stated elsewhere in recent literature as closed-loop (iterated) prediction. Quoting the paper proposed by the reviewer : "Abarbanel, in one of his recent publications (Abarbancl, 1996), revealed the distinction between the dynamic reconstruction and prediction problems. According to him, the capability to solve the prediction problem do not always imply the capability to capture the dynamics of the underlying chaotic system. Dynamic recontruction aims at modeling the attractor dynamics (in state-space) while in the prediction problem only the short term prediction capability is of concern, many of the techniques proposed in the literature for chaotic time-series prediction (see Lillekjendlie et al. (1994) For review) fail at solving the dynamical reconstruction problem (Abarhanel, 1996).
As per the above terminology, the dynamic reconstruction problem may be considered as a system approximation problem, not a function approximation one. This means that the obtained model, though trained in an open loop mode of operation, has to be tested by seeding, at first, its input with a point in the trajectory and, then, feeding back the output to its input to generate recursively the outputs. The reconstructed system should be as close as possible to the original one in tenns of its invariants. Two chaotic systems can be considered to be close not only if they present close short-term evolutions from the same initial condition but also if  their chaotic invariants are sufficiently close. In particular, one cannot consider that a non-chaotic system be a good approximation of a chaotic one.". In our paper, and as stated in the abstract, we are interested in both short-term forecasts and the long-term asymptotic behavior of simulated trajectories (only given the initial condition). The literature proposed by the reviewer 3 only addresses short-term forecasting. The Lorenz-63 dynamics, which lead under specific parameterization as chaotic dynamics, provide a toy example to evaluate both the short-term RMSE and the largest Lyapunov exponent (which is an invariant of the Lorenz 63 dynamical system) as evaluation metrics for learnt models. In the SLA experiment in the other-hand, and due to the lack of stability of the learnt models, we could not compute the Largest lyapunov exponents.  However, we show in the appendices that our model still give realistic forecasts up to 175 days with the same dominant frequency as the true data, which is a significantly larger time horizon compared with the benchmarked  data-driven models.

Overall, we have clarified these aspects in the introduction and related works sections citing the papers suggested by the reviewers and pointing out the differences with the key objectives of our paper.

[1] Mattera & Haykin (1999) "Support vector machines for dynamic reconstruction of a chaotic system"

---

### Decision · Program_Chairs · 2019-12-19

**Decision:**

Reject

**Comment:**

This paper presents an ODE-based latent variable model, argues that extra unobserved dimensions are necessary in general, and that deterministic encodings are also insufficient in general.  Instead, they optimize the latent representation during training.  They include small-scale experiments showing that their framework beats alternatives.

In my mind, the argument about fixed mappings being inadequate is a fair one, but it misses the fact that the variational inference framework already has several ways to address this shortcoming:
1) The recognition network outputs a distribution over latent values, which in itself does not address this issue, but provides regularization benefits.
2) The recognition network is just a strategy for speeding up inference.  There's no reason you can't just do variational inference or MCMC for inference instead (which is similar to your approach), or do semi-amortized variational inference.

Basically, this paper could have been somewhat convincing as a general exploration of approximate inference strategies in the latent ODE model.  Instead, it provides a lot of philosophical arguments and a small amount of empirical evidence that a particular encoder is insufficient when doing MAP inference.  It also seems like a problem that hyperparameters were copied from Chen et al 2018, but are used in a MAP setting instead of a VAE setting.  Finally, it's not clear how hyperparameters such as the size of the latent dimensions were chosen.